# Balanced Training for Sparse GANs

**Yite Wang**[1†], **Jing Wu**[1†], **Naira Hovakimyan**[1], **Ruoyu Sun**[2,3*]
[1]University of Illinois Urbana-Champaign, USA
[2]School of Data Science, The Chinese University of Hong Kong, Shenzhen, China
[3]Shenzhen International Center for Industrial and Applied Mathematics,
Shenzhen Research Institute of Big Data
{yitew2,jingwu6,nhovakim}@illinois.edu, sunruoyu@cuhk.edu.cn

## Abstract

Over the past few years, there has been growing interest in developing larger and deeper neural networks, including deep generative models like generative adversarial networks (GANs). However, GANs typically come with high computational complexity, leading researchers to explore methods for reducing the training and inference costs. One such approach gaining popularity in supervised learning is dynamic sparse training (DST), which maintains good performance while enjoying excellent training efficiency. Despite its potential benefits, applying DST to GANs presents challenges due to the adversarial nature of the training process. In this paper, we propose a novel metric called the balance ratio (BR) to study the balance between the sparse generator and discriminator. We also introduce a new method called balanced dynamic sparse training (ADAPT), which seeks to control the BR during GAN training to achieve a good trade-off between performance and computational cost. Our proposed method shows promising results on multiple datasets, demonstrating its effectiveness. Our code is available at https://github.com/YiteWang/ADAPT.

## 1 Introduction

Generative adversarial networks (GANs) [20, 7, 70, 44] are a type of generative model that has gained significant attention in recent years due to their impressive performance in image-generation tasks. However, the mainstream models in GANs are known to be computationally intensive, making them challenging to train in resource-constrained settings. Therefore, it is crucial to develop methods that can effectively reduce the computational cost of training GANs while maintaining their performance, making GANs more practical and applicable in real-world scenarios.

Neural network pruning has recently emerged as a powerful tool to reduce the training and inference costs of DNNs for supervised learning. There are mainly three genres of pruning methods, namely pruning-at-initialization, pruning-during-training, and post-hoc pruning methods. Post-hoc pruning [32, 42, 25] can date back to the 1980s, which was first introduced for reducing inference time and memory requirements for efficient deployment; hence does not align with our purpose of efficient training. Later, pruning-at-initialization [46, 78, 75] and pruning-during-training methods [84] were introduced to circumvent the need to fully train the dense networks. However, early pruning-during-training algorithms [58] do not bring much training efficiency compared to post-hoc pruning, while pruning-at-initialization methods usually suffer from significant performance drop [18]. Recently, advances in dynamic sparse training (DST) [62, 16, 51, 52, 54] for the first time show that pruning-during-training methods can have comparable training FLOPs as pruning-at-initialization methods

---

*Corresponding author.
[†]Equal contribution

37th Conference on Neural Information Processing Systems (NeurIPS 2023).

while having competing performance to post-hoc pruning. Therefore, applying DST on GANs seems to be a promising choice.

Although DST has attained remarkable achievements in supervised learning, the application of DST on GANs is not successful due to newly emerging challenges. One challenge is keeping the generator and the discriminator balanced. In particular, using overly strong discriminators can lead to overfitting, while weaker discriminators may fail to effectively prevent mode collapse [3, 4]. Hence, balancing the sparse generator and the (possibly) sparse discriminator throughout training is even more difficult. To mitigate the unbalance issue, a recent work STU-GAN [53] proposes to apply DST directly to the generator. However, we find empirically that such an algorithm is likely to fail when the generator is already more powerful than the discriminator. Consequently, it remains unclear how to conduct balanced dynamic sparse training for GANs.

To this end, we propose a metric called balance ratio (BR), which measures the degree of balance of the two components, to study sparse GAN training. We find that BR is useful in (1) understanding the interaction between the discriminator and the generator, (2) identifying the cause of a certain training failure/collapse [7, 8], and (3) helping stabilize sparse GAN training as an indicator. To our best knowledge, this is the first study to quantify the unbalance of sparse GANs and may even provide new insights into dense GAN training.

Furthermore, using BR as an indicator, we propose bAlanced DynAmic sParse Training (`ADAPT`) to adjust the density and the connections of the discriminator automatically during training.

Our main contributions are summarized below:

- We introduce a novel quantity named balance ratio to study the degree of balance in sparse GAN training.
- We find empirically that the balance ratio is problematic in certain practical training scenarios and that existing methods are inadequate for resolving this issue.
- We propose `ADAPT`, which makes real-time monitoring of the balance ratio. By dynamically adjusting the discriminator, `ADAPT` enables effective control of the balance ratio throughout training. Empirically, `ADAPT` achieves a good trade-off between performance and computational cost on several datasets.

## 2 Related works

### 2.1 Neural network pruning

In deep learning, efficiency is achieved through several methods. This paper primarily focuses on model training and inference efficiency, which is different from techniques for data efficiency [83, 85, 86]. These include neural architecture search (NAS) [80, 49] to discover optimal network structures, quantization [31, 67] for computational efficacy, knowledge distillation [28] to leverage the knowledge of larger models for smaller counterparts, and neural network pruning to remove unnecessary connections. Among these, neural network pruning is the focal point of our research. More specifically, we narrow our focus on unstructured pruning [25, 17], where individual weight is the finest resolution. This contracts with structured pruning [56, 59, 57, 30] where entire neurons or channels are pruned.

**Post-hoc pruning.** Post-hoc pruning method prunes weights of a fully-trained neural network. It usually requires high computational costs due to the multiple rounds of the train-prune-retrain procedure [25, 68]. Some use specific criteria [42, 26, 25, 23, 15, 12, 91, 63] to remove weights, while others perform extra optimization iterations [77]. Post-hoc pruning was initially proposed to reduce the inference time, while lottery ticket works [17, 68] aimed to mine trainable sub-networks.

**Pruning-at-initialization methods.** SNIP [46] is one of the pioneering works that aim to find trainable sub-networks without any training. Some follow-up works [78, 75, 13, 65, 1] aim to propose different metrics to prune networks at initialization. Among them, Synflow [75], SPP [45], and FORCE [13] try to address the problem of layer collapse during pruning. NTT [55], PHEW [65], and NTK-SAP [82] draw inspiration from neural tangent kernel theory.

**Pruning-during-training methods.** Another genre of pruning algorithms prunes or adjusts DNNs throughout training. Early works add explicit $\ell_0$ [58] or $\ell_1$ [84] regularization terms to encourage a

sparse solution, hence mitigating performance drop incurred by post-hoc pruning. Later works learn the subnetworks structures through projected gradient descent [94] or trainable masks [73, 88, 35, 41, 50, 71]. However, these pruning-during-training methods often do not introduce memory sparsity during training. As a remedy, DST methods [5, 62, 64, 14, 16, 51, 52, 54, 21] were introduced to train the neural networks under a given parameter budget while allowing mask change during training.

## 2.2 Generative adversarial networks

**Generative adversarial networks (GANs).** GANs [20] have drawn considerable attention and have been widely investigated for years. Deep convolutional GANs [66] replace fully-connected layers in the generator and the discriminator. Follow-up works [22, 36, 7, 92] employed more advanced methods to improve the fidelity of generated samples. After that, several novel loss functions [69, 60, 2, 22, 74], normalization and regularization methods [61, 76, 87] were proposed to stabilize the adversarial training. Besides the efforts devoted to training GANs, image-to-image translation is also extensively explored [96, 95, 10, 38, 43, 81].

**GAN balance.** Addressing the balance between the generator and discriminator in GAN training has been the focus of various works. However, directly applying existing methods to sparse GAN training poses challenges. For instance, [3, 4] offer theoretical analyses on the issue of imbalance but may have limited practical benefits, e.g., they require training multiple generators and discriminators. Empirically, BEGAN [6] proposes to use proportional control theory to maintain a hyper-parameter $\frac{\mathbb{E}[|G(z)-D(G(z))|^\eta]}{\mathbb{E}[|x-D(x)|^\eta]}$, but it is only applicable when the discriminator is an auto-encoder. Unbalanced GAN [24] pretrains a VAE to initialize the generator, which may only address the unbalance near initialization. GCC [48] considers the balance during GAN compression, while its criterion requires a trained (dense) GAN, which is not given in the DST setting. Finally, STU-GAN [53] proposes to use DST to address the unbalance issues but may fail under certain conditions, as demonstrated in our experiments.

**GAN compression and pruning.** One of the promising ways is based on neural architecture search and distillation algorithm [47, 19, 29]. Another part of the work applied pruning-based methods for generator compression [72, 90, 33]. Later, works by [79] presented a unified framework by combing the methods mentioned above. Nevertheless, they only focus on the pruning of generators, thus potentially posing a negative influence on Nash Equilibrium between generators and discriminators. GCC [48] compresses both components of GANs by letting the student GANs also learn the losses. Another line of work [34, 9, 8] tries to test the existence of lottery tickets in GANs. To the best of our knowledge, STU-GAN [53] is the only work that tries to directly train sparse GANs from scratch.

## 3 Preliminary and setups

Generative adversarial networks (GANs) have two fundamental components, a generator $G(\cdot; \boldsymbol{\theta}_G)$ and a discriminator $D(\cdot; \boldsymbol{\theta}_D)$. Specifically, the generator maps a sampled noise $\boldsymbol{z}$ from a multivariate normal distribution $p(\boldsymbol{z})$ into a fake image to cheat the discriminator. In contrast, the discriminator distinguishes the generator's output and the real images $\boldsymbol{x}_r$ from the distribution $p_{\text{data}}(\boldsymbol{x}_r)$.

Formally, the optimization objective of the two-player game defined in GANs can be generally defined as follows:

$$\mathcal{L}_D(\boldsymbol{\theta}_D, \boldsymbol{\theta}_G) = \mathbb{E}_{\boldsymbol{x}_r \sim p_{\text{data}}} \left[ f_1(D(\boldsymbol{x}_r; \boldsymbol{\theta}_D)) \right] + \mathbb{E}_{\boldsymbol{z} \sim p} \left[ f_2(D(G(\boldsymbol{z}; \boldsymbol{\theta}_G))) \right] \tag{1}$$

$$\mathcal{L}_G(\boldsymbol{\theta}_G) = \mathbb{E}_{\boldsymbol{z} \sim p} \left[ g_1(D(G(\boldsymbol{z}; \boldsymbol{\theta}_G))) \right]. \tag{2}$$

To be more specific, different losses can be used, including the loss in the original JS-GAN [20] where $f_1(x) = -\log(x)$, $f_2(x) = -g_1(x) = -\log(1-x)$; Wasserstein loss [22] where $f_1(x) = -f_2(x) = g_1(x) = -x$; and hinge loss [61] where $f_1(x) = \max(0, 1-x)$, $f_2(x) = \max(0, 1+x)$, and $g_1(x) = -x$. The two components are optimized alternately to achieve the Nash equilibrium.

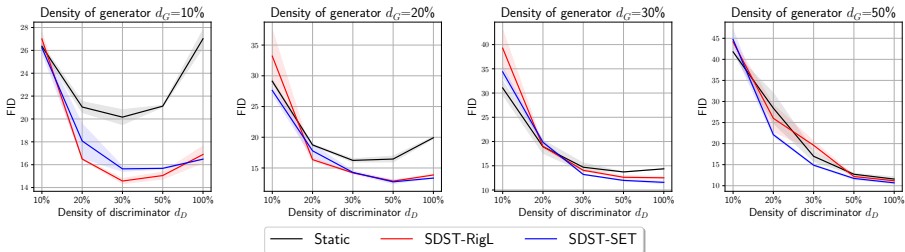

Figure 1: FID ($\downarrow$) comparison of SDST against STATIC sparse training for SNGAN on CIFAR-10 with different sparsity ratio combinations. The shaded areas denote the standard deviation.

**GAN sparse training.** In this work, we are interested in sparse training for GANs. In particular, the objective of sparse GAN training can be formulated as follows:

$$\boldsymbol{\theta}_D^* = \min_{\boldsymbol{\theta}_D} \mathcal{L}_D(\boldsymbol{\theta}_D \odot \boldsymbol{m}_D, \boldsymbol{\theta}_G \odot \boldsymbol{m}_G)$$

$$\boldsymbol{\theta}_G^* = \min_{\boldsymbol{\theta}_G} \mathcal{L}_G(\boldsymbol{\theta}_G \odot \boldsymbol{m}_G)$$

$$\text{s.t.} \quad \boldsymbol{m}_D \in \{0,1\}^{|\boldsymbol{\theta}_D|}, \quad \boldsymbol{m}_G \in \{0,1\}^{|\boldsymbol{\theta}_G|}, \quad \|\boldsymbol{m}_D\|_0/|\boldsymbol{\theta}_D| \leq d_D, \quad \|\boldsymbol{m}_G\|_0/|\boldsymbol{\theta}_G| \leq d_G,$$

where $\odot$ is the Hadamard product; $\boldsymbol{\theta}_D^*$, $\boldsymbol{m}_D$, $|\boldsymbol{\theta}_D|$, $d_D$ are the sparse solution, mask, number of parameters, and target density for the discriminator, respectively. The corresponding variables for the generator are denoted with subscript $G$. For pruning-at-initialization methods, masks $\boldsymbol{m}$ are determined before training, whereas $\boldsymbol{m}$ are dynamically adjusted for dynamic sparse training (DST) methods.

**Dynamic sparse training (DST).** DST methods [62, 16] usually start with a sparse network parameterized by $\boldsymbol{\theta} \odot \boldsymbol{m}$ with randomly initialized mask $\boldsymbol{m}$. After a constant time interval $\Delta T$, it updates mask $\boldsymbol{m}$ by removing a fraction of connections and activating new ones with a certain criterion. The total number of active parameters $\|\boldsymbol{m}\|_0$ is hence kept under a certain threshold $d|\boldsymbol{\theta}|$. Please see Appendix B for more details.

## 4 Motivating observations: The unbalance in sparse GAN training

As discussed in section 1, it is essential to maintain the balance of generator and discriminator during GAN training. As strong discriminators may lead to over-fitting, whereas weak discriminators may be unable to detect mode collapse. When it comes to sparse GAN training, the consequences caused by the unbalance can be further amplified. For example, sparsifying a weak generator while keeping the discriminator unmodified may lead to an even more unbalanced worst-case scenario.

To support our claim, we conduct experiments with SNGAN [61] on the CIFAR-10 dataset. We consider the following sparse training algorithms:

❶ **Static sparse training (STATIC).** For STATIC, layer-wise sparsity ratio and masks $\boldsymbol{m}_G, \boldsymbol{m}_D$ are fixed throughout the training.

❷ **Single dynamic sparse training (SDST).** SDST is a direct application of the DST method on GANs where only the generator dynamically adjusts masks during the training. We name such method SDST as only one component of the GAN, i.e., the generator, is dynamic. Furthermore, we call the variant which grows connections based on gradients magnitude as ● SDST-RigL [16], and randomly as ◆ SDST-SET [62]. Note that STU-GAN [53] is almost identical to ● SDST-RigL with EMA [89] tailored for DST. We do not consider naively applying DST on both generators and discriminators, as in STU-GAN, it is empirically shown that simply adjusting both components generates worse performance with more severe training instability.[2]

We test the considered algorithms with $d_G \in \{10\%, 20\%, 30\%, 50\%\}$ and $d_D \in \{10\%, 20\%, 30\%, 50\%, 100\%\}$. More experiment details can be found at Appendix A and Appendix B.

---

[2]We also perform a small experiment in subsection E.2 to validate their findings.

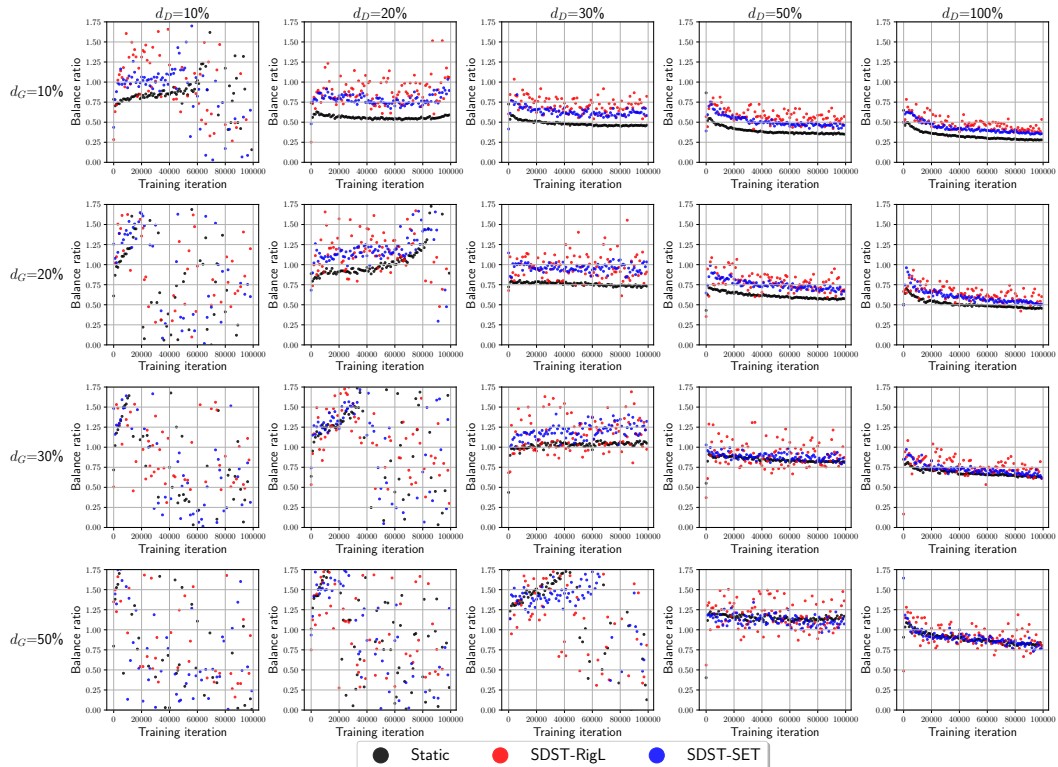

Figure 2: BR comparison of SDST against STATIC sparse training for SNGAN on CIFAR-10 with different sparsity ratio combinations.

### 4.1 Key observations

We report the results in Figure 1 and summarize our critical findings as follows:

❶ **Observation 1: Neither strong nor weak sparse discriminators can provide satisfactory results.** The phenomenon is most noticeable when $d_G = 10\%$, where the FID initially decreases but then increases. The reasons may be as follows: (1) Overly weak discriminators may cause training collapse as they cannot provide useful information to the generator, resulting in a sudden increase in FID at the early stage of sparse GAN training. (2) Overly strong discriminators may not yield good FID results because they learn too quickly, not allowing the generator to keep up. Hence, to ensure a balanced training of GAN for sparse training methods, it is crucial to find an appropriate sparsity ratio for the discriminator.

❷ **Observation 2: SDST is unable to give stable performance boost compared to the STATIC baseline.** Another critical observation is that SDST is better than STATIC only when the discriminator is strong enough. More specifically, for all selected discriminator density ratios, SDST method is not better than STATIC when using a small discriminator density ($d_D = 10\%$). On the contrary, for the cases where $d_D > d_G$, we generally see a significant performance boost brought by SDST.

## 5 Balance ratio: Towards quantifying the unbalance in sparse GAN training

### 5.1 Formulation of the balance ratio

To gain a deeper understanding of the phenomenon observed in the previous section, and to better monitor and control the degree of unbalance in sparse GANs, we introduce a novel quantity called the balance ratio (BR). This quantity is defined as follows.

At each training iteration, we draw random noise $z$ from a multivariate normal distribution and real images $x_r$ from the training set. We denote the discriminator after gradient descent update as $D(\cdot; \theta_D)$. We denote generator before and after gradient descent training as $G^{\text{pre}}(\cdot; \theta_G)$ and

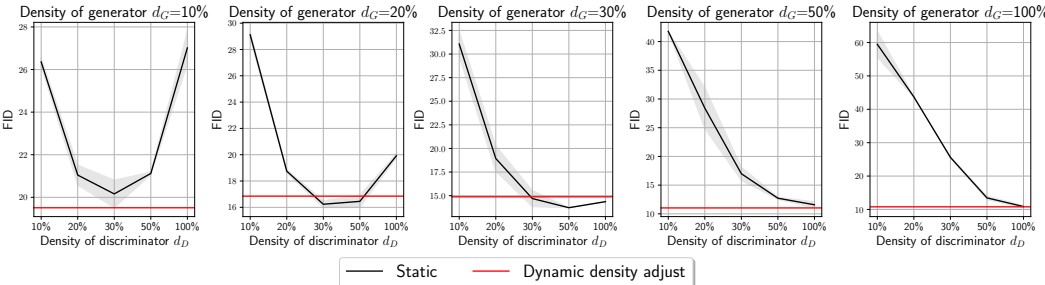

Figure 4: FID ($\downarrow$) of `STATIC` sparsely trained SNGAN with and without `DDA` on CIFAR-10 with different sparsity ratio combinations. The result of `DDA` is independent of $d_D$ as it is determined automatically. The shaded areas denote the standard deviation.

$G^{\text{post}}(\cdot; \boldsymbol{\theta}'_G)$, respectively. Then the balance ratio is defined as:

$$\text{BR} = \frac{\mathbb{E}_{\boldsymbol{z} \sim p}\left[D(G^{\text{post}}(\boldsymbol{z})) - D(G^{\text{pre}}(\boldsymbol{z}))\right]}{\mathbb{E}_{\boldsymbol{x}_r \sim p_{\text{data}}}[D(\boldsymbol{x}_r)] - \mathbb{E}_{\boldsymbol{z} \sim p}[D(G^{\text{pre}}(\boldsymbol{z}))]} = \frac{\alpha}{\beta}. \tag{3}$$

Precisely, BR measures how much improvement the generator can achieve in the scale measured by the discriminator for a specific random noise $\boldsymbol{z}$. When BR is too small (e.g., BR$< 30\%$), the updated generator is too weak to trick the discriminator, as the generated images are still considered fake. Similarly, for the case where BR is too large (e.g., BR$> 80\%$), the discriminator is considered too weak hence it may not provide useful information to the generator. We also illustrate BR in Figure 3.

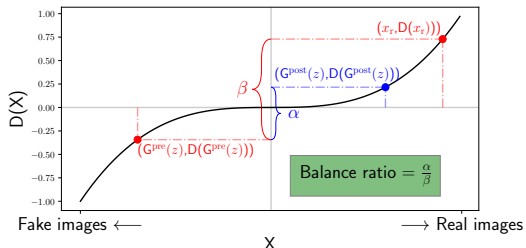

Figure 3: Illustration of balance ratio.

### 5.2 Understanding observation 1: Analysing GAN balance with the balance ratio

We visualize the BR evolution throughout the training for the experiments in section 4 to show the effectiveness of BR in quantifying the balance of sparse GANs. We show the results in Figure 2.

It illustrates that BR can distinguish the density difference (hence the representation power difference) of the discriminator. Specifically, we can see that for larger discriminator density $d_D$, the BR is much lower throughout the training, indicating strong discriminators. On the contrary, for the cases where the discriminators are too weak compared to the generators, e.g., all cases where $d_D = 10\%$, we can observe BR first increases and then oscillates wildly. We believe this oscillatory behavior is related to the training collapse. Empirical results also show that the FID metric experiences a sudden increase after this turning point.

### 5.3 Dynamic density adjust: A first attempt to utilize the balance ratio

As demonstrated in the previous section, the balance ratio (BR) effectively captures the degree of balance between the generators and discriminators in sparse GANs. Hence, it is natural to leverage BR to dynamically adjust the density of discriminators during sparse GAN training so that a reasonable discriminator density can be found.

To demonstrate the value of BR, we propose a simple yet powerful modification to the `STATIC` baseline. This method, which we call **dynamic density adjust (`DDA`)**, is explained below. Specifically, we initialize the initial density of the discriminator $d_D^{\text{init}} = d_G$. After a specific training iteration interval, we adjust the density of the discriminator based on the BR over the last few iterations with a pre-defined density increment $\Delta d$. With a pre-defined BR bounds $[B_-, B_+]$, we decrease $d_D$ by $\Delta d$ when BR is smaller than $B_-$, and vise versa. We show the algorithm in Appendix C Algorithm 1.

**Comparison to ADA [37].** In this paragraph, we compare ADA and `DDA`. (1) Notice that `DDA` algorithm is orthogonal to ADA in a sense that StyleGAN2-ADA adjusts the data augmentation

probability while `DDA` adjusts the discriminator density. (2) Moreover, the criterion used in `DDA`, i.e. BR, is very different from the criterion proposed in StyleGAN2-ADA, i.e. $r_v = \frac{\mathbb{E}[D(x_{\text{train}})] - \mathbb{E}[D(x_{\text{val}})]}{\mathbb{E}[D(x_{\text{train}})] - \mathbb{E}[D(G(z))]}$ and $r_t = \mathbb{E}[\text{sign}(D(x_{\text{train}}))]$. In particular, $r_v$ requires a separate validation set, while $r_t$ only quantifies the overfitting of the discriminator to the training set. (3) Another note is that `DDA` is a flexible framework, where its criterion, i.e. BR, can be potentially replaced by $r_v$, $r_t$, and such.

**Experiment results.** We test `DDA` with target BR interval $[B_-, B_+] = [0.5, 0.65]$. Precisely, `DDA` tends to find a suitable discriminator where the generator can just trick the discriminator throughout the training. We show the results in Figure 4 with red lines. The experiments show that `DDA` can identify reasonable discriminator densities to enable balanced training for sparse GANs.

### 5.4 Understanding observation 2: Analysing the failure of `SDST` with the balance ratio

By leveraging BR, we can also gain further insights into why some configurations do not benefit from `SDST` as compared to `STATIC`.

**Regarding `SDST` as a way of increasing the generator capacity.** Our findings suggest that `SDST` can possibly enhance the generator's representation power, as demonstrated by the higher BR values compared to `STATIC` observed in Figure 2. We attribute this effect to the in-time over-parameterization (ITOP) [54] induced by dynamic sparse training.

**`SDST` does not address training collapse.** The increase in generator's representation power resulting from `SDST` is only beneficial when the discriminator has matching or superior representation power. Therefore, if the training has already collapsed for the static baseline methods (`STATIC`), meaning that the generator is already stronger than the discriminator, `SDST` may not be effective in stabilizing sparse GAN training. This is evident from the results shown in Figure 2 first-row column 1, second-row column 1, third-row columns 1-2, and fourth-row columns 1-3.

Despite the superior performance of STU-GAN (or `SDST` in general) at higher discriminator density ratios $d_D$, there exist some limitations for `SDST`, which we summarize below:

❶ `SDST` requires a pre-defined discriminator density $d_D$ before training. However, it is unclear what is a good choice. In real-world scenarios, it is not practical to manually search for the optimal $d_D$ for each $d_G$. A workaround may be using the maximum allowed density for the discriminator. However, as shown in Figure 1, the best performance is not always obtained with the maximum $d_D = 100\%$. Moreover, we are wasting extra computational cost for a worse performance if we use an overly-strong discriminator.

❷ `SDST` fails if there is an additional constraint on the density of the discriminator $d_D$. As Figure 1 suggests, for weak discriminators, `SDST` is unable to show consistent improvement compared to the `STATIC` baseline.

Hence, STU-GAN (or `SDST` in general), which directly applies DST to the generator, may only be useful when the corresponding discriminator is strong enough. In this sense, obtaining balanced training automatically is essential in GAN DST to deal with more complicated scenarios.

## 6 Balanced dynamic sparse training for GANs

In this section, we describe our methodology for balanced sparse GAN training.

STU-GAN (or `SDST` in general) considered in the last section cannot generate stable and satisfying performance. This implies that we should utilize the discriminator in a better way rather than do nothing (like `SDST`) or directly apply DST to the discriminator (see subsection E.2 for additional experiments). Consequently, `DDA` (subsection 5.3), which adjusts the discriminator density to stabilize GAN training, is a favorable candidate to address the issue. To this end, we propose **bAlanced DynAmic sParse Training (`ADAPT`)**, which adjusts the density of the discriminator during training with `DDA` while the generator performs DST.

We further introduce two variants, namely $\text{ADAPT}_{\text{relax}}$ and $\text{ADAPT}_{\text{strict}}$, based on whether we force the discriminator to be sparse. We present them in subsection 6.1 and subsection 6.2. These methods are more flexible and generate more stable performance compared to `SDST`.

Table 1: FID ($\downarrow$) of different sparse training methods **with no constraint on the density of the discriminator**. Best results are in **bold**; second-best results are underlined.

| Dataset | CIFAR-10 (SNGAN) | | | | STL-10 (SNGAN) | | | | CIFAR-10 (BigGAN) | | | | TinyImageNet (BigGAN) | | | |
|---|---|---|---|---|---|---|---|---|---|---|---|---|---|---|---|---|
| Generator density | 10% | 20% | 30% | 50% | 10% | 20% | 30% | 50% | 10% | 20% | 30% | 50% | 10% | 20% | 30% | 50% |
| (Dense Baseline) | 10.74 | | | | 29.71 | | | | 8.11 | | | | 15.43 | | | |
| STATIC-Balance | 26.75 | 19.04 | 15.05 | 12.24 | 48.18 | 44.67 | 41.73 | 37.68 | 16.98 | 12.81 | 10.33 | 8.47 | 28.78 | 21.67 | 18.86 | 17.51 |
| STATIC-Strong | 26.79 | 19.65 | 14.38 | 11.91 | 52.48 | 43.85 | 42.06 | 37.47 | 23.48 | 14.26 | 11.19 | 8.64 | 31.44 | 22.51 | 18.22 | 18.00 |
| ◆ SDST-Balance-SET | 26.23 | 17.79 | 13.21 | 11.79 | 56.41 | 46.58 | 39.93 | 30.37 | 12.41 | 9.87 | 9.13 | **8.01** | 25.39 | 21.30 | 21.80 | 21.20 |
| ◆ SDST-Strong-SET | 16.49 | 13.36 | **11.68** | 10.68 | 67.37 | 49.96 | 37.99 | 31.08 | 18.94 | 9.64 | 8.75 | 8.36 | 22.20 | 20.56 | 21.70 | 18.32 |
| ● SDST-Balance-RigL | 27.06 | 16.36 | 14.00 | 12.28 | 43.08 | 33.90 | 31.83 | 30.30 | 12.45 | 9.42 | 8.86 | 8.03 | 21.60 | 19.33 | 18.57 | 17.45 |
| ● SDST-Strong-RigL | 17.02 | 13.86 | 12.51 | 11.35 | 53.65 | 33.25 | **31.41** | 30.18 | 10.58 | 9.11 | 8.69 | 8.33 | 21.14 | 18.95 | 17.75 | 16.30 |
| ADAPT$_{\text{relax}}$ (Ours) | **14.19** | **13.19** | 12.38 | **10.60** | **35.98** | **33.06** | 31.71 | **29.96** | **10.19** | **8.56** | **8.36** | 8.22 | **19.42** | **17.99** | **17.06** | **14.15** |

## 6.1 ADAPT$_{\text{relax}}$: Balanced dynamic sparse training in the relaxed setting

In this section, we consider the relaxed setting where a dense discriminator can be used, i.e., $d_D \leq d_{\max} = 100\%$. This relaxed scenario gives the greatest flexibility to the discriminator. However, it does not necessarily enforce the sparsity of the discriminator (hence, no computational savings for the discriminator) because the density of the discriminator can be as high as $100\%$.

For the relaxed setting, we use the direct combination of SDST with DDA. Precisely, the generator is adjusted using DST as mentioned in section 4 while the density of the discriminator is dynamically adjusted with DDA as mentioned in subsection 5.3. We call such a combination **relaxed balanced dynamic sparse training (ADAPT$_{\text{relax}}$)**. Please see Appendix C Algorithm 2 for more details.

**Comparison to STU-GAN (or SDST in general)**. Compared to STU-GAN (or SDST in general), which pre-defines and fixes the discriminator density during training, the difference is that for ADAPT$_{\text{relax}}$, the density of the discriminator is adjusted during the training process automatically through real-time monitoring of the balance ratio. Given the initial discriminator density $d_D^{\text{int}} = d_G$, ADAPT$_{\text{relax}}$ increases the discriminator density if a stronger discriminator is needed, and vice versa.

## 6.2 ADAPT$_{\text{strict}}$: Balanced dynamic sparse training in the strict setting

Different from subsection 6.1, we now consider a strict setting where there is an additional sparsity constraint on the discriminator density in this section, i.e., $d_D \leq d_{\max} < 100\%$.

ADAPT$_{\text{relax}}$ introduced in the previous section does not necessarily enforce sparsity for the discriminator, which provides less memory/training resources saving for larger generator density ratios. Note that the discriminator does not take advantage of DST to explore the structure of the dense network. Hence, we further present **strict balanced dynamic sparse training (ADAPT$_{\text{strict}}$)** in this section. This method allows the discriminator to perform DST in a controlled manner, which can lead to a better exploration of the dense network structure while maintaining the balance between the generator and the discriminator. We explain how ADAPT$_{\text{strict}}$ differs from ADAPT$_{\text{relax}}$ below:

❶ **Capacity increase of the discriminator.** The essential difference lies when the observed BR is higher than $B_+$, which means we need a stronger discriminator. In this case, if the discriminator density is lower than the constraint, i.e., $d_D < d_{\max}$, ADAPT$_{\text{strict}}$ will perform just like ADAPT$_{\text{relax}}$ to increase the discriminator density. However, if the discriminator is already the maximum density, i.e., $d_D = d_{\max}$, the discriminator will alternatively perform DST as a way of increasing the discriminator capacity (See subsection 5.4 for intuition).

❷ **Capacity decrease of the discriminator.** Similar to ADAPT$_{\text{relax}}$, when the observed BR is lower than $B_-$, we will decrease the discriminator density.

Hence, ADAPT$_{\text{strict}}$ makes the discriminator adaptive both in the density level (through density adjustment) and the parameter level (through DST). The algorithm of ADAPT$_{\text{strict}}$ is shown in Appendix C Algorithm 3.

## 6.3 Experiment setting

**Datasets, architectures, and target sparsity ratios.** We conduct experiments on SNGAN with ResNet architectures on the CIFAR-10 [40] and the STL-10 [11] datasets. We have also conducted experiments with BigGAN [7] on the CIFAR-10 and TinyImageNet dataset (with DiffAug [93]).

Table 2: FID ($\downarrow$) of different sparse training methods. **The density of the discriminator is constrained to be lower than 50%**. Best results are in **bold**; second-best results are underlined.

| Dataset | CIFAR-10 (SNGAN) | | | | STL-10 (SNGAN) | | | | CIFAR-10 (BigGAN) | | | | TinyImageNet (BigGAN) | | | |
|---|---|---|---|---|---|---|---|---|---|---|---|---|---|---|---|---|
| Generator density | 10% | 20% | 30% | 50% | 10% | 20% | 30% | 50% | 10% | 20% | 30% | 50% | 10% | 20% | 30% | 50% |
| (Dense Baseline) | 10.74 | | | | 29.71 | | | | 8.11 | | | | 15.43 | | | |
| STATIC-Balance | 26.75 | 19.04 | 15.05 | 12.58 | 48.18 | 44.67 | 41.73 | 37.68 | 16.98 | 12.81 | 10.33 | 8.47 | 28.78 | 21.67 | 18.86 | 17.51 |
| STATIC-Strong | 21.73 | 16.69 | 13.48 | 12.58 | 50.36 | 44.06 | 40.73 | 37.68 | 18.91 | 13.43 | 10.84 | 8.47 | 33.01 | 23.93 | 17.90 | 17.51 |
| ◆ SDST-Balance-SET | 26.23 | 17.79 | 13.21 | **11.79** | 56.24 | 44.51 | 41.23 | 30.80 | 12.41 | 9.87 | 9.13 | 8.01 | 25.39 | 21.30 | 21.80 | 21.20 |
| ◆ SDST-Strong-SET | 15.68 | 12.75 | **11.98** | **11.79** | 57.91 | 50.05 | 38.13 | 30.80 | 11.85 | 9.39 | 8.61 | 8.01 | 22.68 | 20.24 | 22.00 | 21.20 |
| ● SDST-Balance-RigL | 27.06 | 16.36 | 14.00 | 12.28 | 43.08 | 33.90 | 31.64 | 30.30 | 12.45 | 9.42 | 8.86 | 8.03 | 21.60 | 19.33 | 18.57 | 17.45 |
| ● SDST-Strong-RigL | 15.19 | 12.93 | 12.75 | 12.28 | 53.74 | 37.34 | 31.98 | 30.30 | 10.11 | 9.17 | **8.35** | 8.03 | 21.90 | 20.43 | 18.29 | 17.45 |
| ADAPT$_{\text{strict}}$ (Ours) | **14.53** | **12.73** | 12.20 | 12.11 | **41.18** | **31.59** | **31.16** | **29.11** | **9.29** | **8.64** | 8.44 | **7.90** | **18.89** | **17.37** | **16.93** | **16.02** |

Table 3: Normalized training FLOPs ($\downarrow$) of different sparse training methods with no constraint on the density of the discriminator.

| Dataset | CIFAR-10 (SNGAN) | | | | STL-10 (SNGAN) | | | | CIFAR-10 (BigGAN) | | | | TinyImageNet (BigGAN) | | | |
|---|---|---|---|---|---|---|---|---|---|---|---|---|---|---|---|---|
| Generator density | 10% | 20% | 30% | 50% | 10% | 20% | 30% | 50% | 10% | 20% | 30% | 50% | 10% | 20% | 30% | 50% |
| (Dense Baseline) | 100% ($2.67 \times 10^{17}$) | | | | 100% ($3.94 \times 10^{17}$) | | | | 100% ($6.81 \times 10^{17}$) | | | | 100% ($9.85 \times 10^{17}$) | | | |
| Static-Balance | 8.97% | 17.08% | 26.25% | 47.25% | 27.30% | 47.14% | 59.22% | 73.35% | 9.79% | 19.02% | 28.66% | 49.03% | 23.25% | 44.87% | 60.91% | 79.29% |
| Static-Strong | 58.29% | 60.94% | 64.53% | 74.61% | 86.12% | 86.94% | 87.60% | 88.84% | 83.78% | 84.80% | 86.21% | 90.15% | 48.02% | 61.62% | 72.79% | 85.79% |
| ◆ SDST-Balance-SET | 9.78% | 18.91% | 28.35% | 48.44% | 27.55% | 47.60% | 60.17% | 75.38% | 10.35% | 20.12% | 29.96% | 49.82% | 21.13% | 37.06% | 48.83% | 65.58% |
| ◆ SDST-Strong-SET | 59.25% | 62.94% | 66.89% | 75.96% | 86.36% | 87.43% | 88.49% | 90.82% | 84.36% | 85.90% | 87.52% | 90.95% | 45.66% | 53.91% | 60.61% | 71.88% |
| ● SDST-Balance-RigL | 10.71% | 17.43% | 25.66% | 43.56% | 29.51% | 50.41% | 63.34% | 79.03% | 9.92% | 19.30% | 28.90% | 48.31% | 24.97% | 43.86% | 57.26% | 76.75% |
| ● SDST-Strong-RigL | 58.63% | 61.35% | 64.04% | 71.01% | 88.51% | 90.24% | 91.78% | 94.57% | 83.97% | 85.24% | 86.59% | 89.54% | 50.05% | 61.02% | 69.64% | 83.35% |
| ADAPT$_{\text{relax}}$ (Ours) | 36.67% | 57.62% | 61.31% | 70.11% | 46.73% | 77.92% | 83.62% | 90.49% | 10.39% | 25.90% | 40.65% | 80.76% | 29.75% | 51.98% | 64.57% | 80.81% |

Target density ratios of the generators $d_G$ are chosen from $\{10\%, 20\%, 30\%, 50\%\}$. Please see Appendix A for more experiment details.

**Baseline methods and two practical strategies.** We use STATIC and SDST (section 4) as our baselines. Note that in real-world application scenarios, it is not practical to perform a grid search for a good $d_D$ as in section 4. Hence, we propose two practical strategies to define the constant discriminator density for these baseline methods: (1) balance strategy, where we set the density of the discriminator $d_D$ the same as the density of the generator $d_G$; (2) strong strategy, where we set the density of the discriminator as large as possible, i.e., $d_D = d_{\max}$. For SDST methods, we test both ◆ SDST-SET and ● SDST-RigL. For a fair comparison, $d_{\max}$ is set to be 100% and 50% for the relaxed setting and the strict setting, respectively.

For ADAPT, we use the ● RigL version, which grows connections of the generators and discriminators based on gradient magnitude. The gradient information enables two components to react promptly according to the change of each other. Different from the value used in subsection 5.3, we control the balance ratio in the range $[0.45, 0.55]$ unless otherwise mentioned to have a slightly stronger discriminator, potentially avoiding training collapse. More details can be found in Appendix B.

### 6.4 Experiment results

We show the experiment results in Table 1 and Table 2 for the relaxed setting and the strict setting, respectively. We also present the training FLOPs normalized by the dense counterpart for the relaxed setting in Table 3. We defer the results for the strict setting to Appendix F Table 12. We show FID for the CIFAR-10 test set, Inception scores, and comparison with post-hoc pruning baseline in Appendix E. We also show ADAPT BR evolution in Appendix D. We summarize our findings below.

**The strong strategy and the balance strategy for baselines.** Generally, using the strong strategy has some advantages over the balance strategy. Such an observation is most prominent in the CIFAR-10 dataset. For the cases where the balance strategy is better, e.g., SNGAN on the STL-10 dataset, our explanation is that the size difference between generators and discriminators is more significant. Hence, the degree of unbalance is more severe and leads to more detrimental effects.

**Comparison of ● RigL and ◆ SET for SDST.** We found that ● RigL has an advantage over ◆ SET when dealing with more sparse generators. Our hypothesis is that gradient information can effectively guide the generator to identify the most crucial connections in such cases. However, this advantage is not as apparent for more dense generators.

**ADAPT$_{\text{relax}}$ achieves a good trade-off between performance and computational cost.** Experiments show that ADAPT$_{\text{relax}}$ shows promising performance by being best for 13 out of 16 cases. The advantage of ADAPT$_{\text{relax}}$ is most prominent for the most difficult case, i.e., $d_G = 10\%$. Specifically, it

shows 2.3 and 7.1 FID improvements over the second-best methods for the SNGAN on the CIFAR-10 and the STL-10, respectively. Moreover, compared to the competitive baseline methods that use the strong strategy, i.e., ● `SDST-Strong-RigL` and ◆ `SDST-Strong-SET`, ADAPT$_{relax}$ shows great computational cost reduction. For example, it outperforms ● `SDST-Strong-RigL` on BigGAN (CIFAR-10) with much-reduced training FLOPs (10.39% v.s. 83.97%).

**ADAPT$_{strict}$ shows stable and superior performance.** Similar to ADAPT$_{relax}$, we notice that ADAPT$_{strict}$ also delivers promising results compared to baselines, even with a further constraint on the discriminator. More precisely, among all the cases, ADAPT$_{strict}$ ranks top 2 for all cases, with 13 cases being the best. Moreover, ADAPT$_{strict}$ again shows comparable or better performance compared to ● `SDST-Strong-RigL` with reduced computational cost.

A more interesting observation is that ADAPT$_{strict}$ sometimes outperforms ADAPT$_{relax}$. We speculate that this phenomenon occurs because changes in density may result in a larger influence on the GAN balance during training compared to DST. Hence, the strict version, whose discriminator density range is smaller, may offer a more consistent performance.

# 7 Conclusion

In this paper, we investigate the use of DST for GANs and find that solely applying DST to the generator does not necessarily enhance the performance of sparse GANs. To address this, we introduce BR to examine the degree of unbalance between the sparse generators and discriminators. We find that applying DST to the generator only benefits the training when the discriminator is comparatively stronger. Additionally, we propose ADAPT, which can dynamically adjust the discriminator at both the parameter and density levels. Our approach shows promising results, and we hope it can aid researchers in better comprehending the interplay between the two components of GAN training and motivate further exploration of sparse training for GANs. However, we must note that we have not yet evaluated our methods on the latest GAN architectures due to computational constraints.

# 8 Acknowledgement

This work utilizes resources supported by the National Science Foundation's Major Research Instrumentation program, grant No.1725729 [39], as well as the University of Illinois at Urbana-Champaign. This work is supported in part by Hetao Shenzhen-Hong Kong Science and Technology Innovation Cooperation Zone Project (No.HZQSWS-KCCYB-2022046); University Development Fund UDF01001491 from the Chinese University of Hong Kong, Shenzhen; Guangdong Key Lab on the Mathematical Foundation of Artificial Intelligence, Department of Science and Technology of Guangdong Province. Prof. NH is in part supported by Air Force Office of Scientific Research (AFOSR) grant FA9550-21-1-0411.

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

## Overview of the Appendix

The Appendix is organized as follows:

- Appendix A introduces the general experimental setup.
- Appendix B introduces the details of dynamic sparse training.
- Appendix C shows detailed algorithms, i.e., `DDA`, `ADAPT`$_{\text{relax}}$, and `ADAPT`$_{\text{strict}}$.
- Appendix D shows the BR evolution during training for `ADAPT`.
- Appendix E shows additional results, including IS and FID of test sets of the main paper.
- Appendix F shows detailed FLOPs comparisons of sparse training methods.

## A Experimental setup

In this section, we explain the training details used in our experiments. Our code is mainly based on the original code of ITOP [54] and GAN ticket [9].

### A.1 Architecture details

We use ResNet-32 [27] for the CIFAR-10 dataset and ResNet-48 for the STL-10 dataset. See Table 4 and Table 5 for detailed architectures. We apply spectral normalization for all fully-connected layers and convolutional layers of the discriminators.

For BigGAN architecture, we use the implementation used in DiffAugment [93].[3]

### A.2 Datasets

We use the training set of CIFAR-10, the unlabeled partition of STL-10, and the training set of TinyImageNet for GAN training. Training images are resized to $32 \times 32$, $48 \times 48$, $64 \times 64$ for CIFAR-10, STL-10, and TinyImageNet datasets, respectively. Augmentation methods for both datasets are random horizontal flip and per-channel normalization.

### A.3 Training hyperparameters

**SNGAN on the CIFAR-10 and STL-10 datasets.** We use a learning rate of $2 \times 10^{-4}$ for both generators and discriminators. The discriminator is updated five times for every generator update. We adopt Adam optimizer with $\beta_1 = 0$ and $\beta_2 = 0.9$. The batch size of the discriminator and the generator is set to 64 and 128, respectively. Hinge loss is used following [7, 9]. We use exponential moving average (EMA) [89] with $\beta = 0.999$. The generator is trained for a total of 100k iterations.

**BigGAN on the CIFAR-10 dataset.** We use a learning rate of $2 \times 10^{-4}$ for both generators and discriminators. The discriminator is updated four times for every generator update. We adopt Adam optimizer with $\beta_1 = 0$ and $\beta_2 = 0.999$. The batch size of both the discriminator and the generator is set to 50. Hinge loss is used following [7, 87]. We use EMA with $\beta = 0.9999$. The generator is trained for a total of 200k iterations.

**BigGAN on the TinyImageNet dataset.** We use DiffAug [93] to augment the input. The learning rate of the discriminator and the generator are set to $4 \times 10^{-4}$ and $1 \times 10^{-4}$, respectively. The discriminator is updated one time for every generator update. We adopt Adam optimizer with $\beta_1 = 0$ and $\beta_2 = 0.999$. The batch size of both the discriminator and the generator is set to 256. Hinge loss is used following [7, 87]. We use EMA with $\beta = 0.9999$. The generator is trained for a total of 200k iterations.

### A.4 Evaluation metric

**SNGAN on the CIFAR-10 and the STL-10 datasets.** We compute Fréchet inception distance (FID) and Inception score (IS) for 50k generated images every 5000 iterations. Best FID and IS are

---

[3] `https://github.com/mit-han-lab/data-efficient-gans/tree/master/DiffAugment-biggan-cifar`.

reported. For the CIFAR-10 dataset, we report both FID for the training set and test set, whereas, for the STL-10 dataset, we report the FID of the unlabeled partition.

**BigGAN on the CIFAR-10 and the TinyImageNet dataset.** We compute Fréchet inception distance (FID) and Inception score (IS) for 10k generated images every 5000 iterations. Best FID and IS are reported.

# B  Dynamic sparse training details

## B.1  How the generator performs DST

In this section, we explain how the generator performs DST below. Note that the generator performs the same for `SDST` and `ADAPT`.

**Sparsity distribution at initialization.** Following RigL and ITOP [16, 54], only parameters of fully connected and convolutional layers will be pruned. At initialization, we use the commonly adopted *Erdős-Rényi-Kernel* (ERK) strategy [16, 14, 54] to allocate higher sparsity to larger layers. Specifically, the sparsity of convolutional layers $l$ is scaled with $1 - \frac{n^{l-1}+n^l+w^l+h^l}{n^{l-1}n^l w^l h^l}$, where $n^l$ denotes the number of channels of layer $l$ while $w^l$ and $h^l$ are the widths and the height of the corresponding kernel in that layer. For fully connected layers, *Erdős-Rényi* (ER) strategy is used, where the sparsity is scaled with $1 - \frac{n^{l-1}+n^l}{n^{l-1}n^l}$.

**Update schedule.** The update schedule controls how many connections are adjusted per DST operation. It can be specified by the number of training iterations between sparse connectivity updates $\Delta T_G$, the initial fraction of connections adjusted $\gamma$, and decaying schedule $f_{\text{decay}}(\gamma, T)$ for $\gamma$.

**Drop and grow.** After $\Delta T_G$ training iterations, we update the mask $\boldsymbol{m}_G$ by dropping/pruning $f_{\text{decay}}(\gamma, T) |\boldsymbol{\theta}_G| d_G$ number of connections with the lowest magnitude, where $|\boldsymbol{\theta}_G|$, $d_G$ are the number of parameters and target density for the generator, $f_{\text{decay}}(\gamma, T)$ is the update schedule. Right after the connection drop, we regrow the same amount of connections.

For the growing criterion, we test both random growth ◆ SET [62, 54] and gradient-based growth ● RigL [16]. Concretely, gradient-based methods find newly-activated connections $\theta$ with the highest gradient magnitude $\left|\frac{\partial \mathcal{L}}{\partial \theta}\right|$, while random-based methods explore connections in a random fashion. All the newly-activated connections are set to 0. One thing that should be noticed is that while previous works consider layer-wise connections drop and growth, we grow and drop connections globally as it grants more flexibility to the DST method.

**EMA for sparse GAN.** EMA [89] is well-known for its ability to alleviate the non-convergence of GAN. We also implement EMA for sparse GAN training. Specifically, we zero out the moving average of dropped weights whenever there is a mask change.

## B.2  DST hyperparameters for the generator

We use the same hyper-parameters for `SDST` and `ADAPT`. The initial $\gamma$ is set to 0.5, and we use a cosine annealing function $f_{\text{decay}}$ following RigL and ITOP.

**SNGAN on the CIFAR-10 and the STL-10 datasets.** The connection update frequency of the generator $\Delta T_G$ is set to 500 and 1000 for the CIFAR-10 dataset and STL-10 dataset, respectively.

**BigGAN on the CIFAR-10 and the TinyImageNet dataset.** The connection update frequency of the generator $\Delta T_G$ is set to be 1000.

## B.3  Density dynamic adjust (`DDA`) hyper-parameters

In this section, we provide hyper-parameters used in subsection 5.3. We set $d_D = 2000$, $\Delta T_D = 0.05$, $[B_-, B_+] = [0.5, 0.65]$. Time-averaged BR over 1000 iterations is used as the indicator.

## B.4  DST hyperparameters for the discriminator in `ADAPT`

We use a constant BR interval $[B_-, B_+] = [0.45, 0.55]$ for SNGAN experiments and BigGAN on the CIFAR-10 dataset. We set the BR interval $[B_-, B_+] = [0.3, 0.4]$ for BigGAN on the TinyImageNet

Table 4: ResNet architecture for CIFAR-10.

| (a) Generator | (b) Discriminator |
|---|---|
| $z \in \mathbb{R}^{128} \sim \mathcal{N}(0, I)$ | image $x \in [-1, 1]^{32 \times 32 \times 3}$ |
| dense, $4 \times 4 \times 256$ | ResBlock down 128 |
| ResBlock up 256 | ResBlock down 128 |
| ResBlock up 256 | ResBlock down 128 |
| ResBlock up 256 | ResBlock down 128 |
| BN, ReLU, $3 \times 3$ conv, Tanh | ReLU 0.1 |
| | Global sum pooling |
| | dense $\to 1$ |

Table 5: ResNet architecture for STL-10.

| (a) Generator | (b) Discriminator |
|---|---|
| $z \in \mathbb{R}^{128} \sim \mathcal{N}(0, I)$ | image $x \in [-1, 1]^{48 \times 48 \times 3}$ |
| dense, $6 \times 6 \times 512$ | ResBlock down 64 |
| ResBlock up 256 | ResBlock down 128 |
| ResBlock up 128 | ResBlock down 256 |
| ResBlock up 64 | ResBlock down 512 |
| BN, ReLU, $3 \times 3$ conv, Tanh | ResBlock down 1024 |
| | ReLU 0.1 |
| | Global avg pooling |
| | dense $\to 1$ |

since it uses DiffAug. Time-averaged BR over 1000 iterations is used as the indicator. Density increment $\Delta d$ is set to be 0.05, 0.025, and 0.05 for SNGAN (CIFAR-10), SNGAN (STL-10), and BigGAN (CIFAR-10), respectively. We use the same setting used in subsection B.2 for the generator.

**Hyper-parameters for ADAPT_relax.** The density update frequency of the discriminator $\Delta T_D$ is 1000, 2000, 5000, and 10000 iterations for SNGAN (CIFAR-10), SNGAN (STL-10), BigGAN (CIFAR-10), and BigGAN (TinyImageNet), respectively.

**Hyper-parameters for ADAPT_strict.** The density/connections update frequency of the discriminator $\Delta T_D$ is 2000, 2000, 5000, and 10000 iterations for SNGAN (CIFAR-10), SNGAN (STL-10), BigGAN (CIFAR-10), and BigGAN (TinyImageNet), respectively.

Note that we compute BR for every iteration to visualize the BR evolution, whereas one should note that such computational cost can be greatly decreased if BR is computed every few iterations.

## C   Algorithms

In this section, we present the detailed algorithms for DDA, ADAPT_relax, and ADAPT_strict.

### C.1   Dynamic adjust algorithm

We first present DDA in Algorithm 1.

---

**Algorithm 1** Dynamic density adjust (DDA) for the discriminator.

---

**Require:** Generator $G$, discriminator $D$, BR upper bound $B_+$ and lower bound $B_-$, DA interval $\Delta T_D$, density increment $\Delta d$, current training iteration $t$.
1: **if** $t \mod \Delta T_D == 0$ **then**
2:     Compute time-averaged BR with Equation 3
3:     **if** BR $> B_+$ **then**
4:         Increase the density of discriminator from $d_D$ to $d_D + \Delta d$.
5:     **else if** BR $< B_-$ **then**
6:         Decrease the density of discriminator from $d_D$ to $d_D - \Delta d$.
7:     **end if**
8: **end if**

---

### C.2   Relaxed balanced dynamic sparse training algorithm

Details of ADAPT_relax algorithm is presented in Algorithm 2.

### C.3   Strict balanced dynamic sparse training algorithm

Details of ADAPT_strict algorithm is presented in Algorithm 3.

**Algorithm 2** Relaxed balanced dynamic sparse training ($\texttt{ADAPT}_{\text{relax}}$) for GANs.

---

**Require:** Generator $G$, discriminator $D$, total number of training iterations $T$, number of training steps for
     discriminator in each iteration $N$, discriminator adjustment interval $\Delta T_D$, DST interval for the generator
     $\Delta T_G$, density increment $\Delta d$, target generator density $d_G$, BR upper bound $B_+$ and lower bound $B_-$.
 1: Set initial discriminator density $d_D = d_G$
 2: **for** $t$ in $[1, \cdots, T]$ **do**
 3:    **for** $n$ in $[1, \cdots, N]$ **do**
 4:       Compute the loss of discriminator $\mathcal{L}_D(\boldsymbol{\theta}_D)$
 5:       $\mathcal{L}_D(\boldsymbol{\theta}_D).backward()$
 6:    **end for**
 7:    **if** $t \mod \Delta T_D == 0$ **then**
 8:       Compute the loss of generator $\mathcal{L}_G(\boldsymbol{\theta}_G)$
 9:       $\mathcal{L}_G(\boldsymbol{\theta}_G).backward()$
10:       Compute time-averaged BR with Equation 3
11:       **if** BR $> B_+$ **then**
12:          Increase the density of discriminator from $d_D$ to $\min(100\%, d_D + \Delta d)$.
13:       **else if** BR $< B_-$ **then**
14:          Decrease the density of discriminator from $d_D$ to $\max(0\%, d_D - \Delta d)$.
15:       **end if**
16:    **end if**
17:    **if** $t \mod \Delta T_G == 0$ **then**
18:       Apply DST to $G$
19:    **end if**
20: **end for**

---

**Algorithm 3** Strict balanced dynamic sparse training ($\texttt{ADAPT}_{\text{strict}}$) for GANs.

---

**Require:** Generator $G$, discriminator $D$, total number of training iterations $T$, number of training steps for
     discriminator in each iteration $N$, given maximal density of discriminator $d_{\max}$, discriminator adjustment
     interval $\Delta T_D$, DST interval for the generator $\Delta T_G$, density increment $\Delta d$, target generator density $d_G$, BR
     upper bound $B_+$ and lower bound $B_-$.
 1: Set initial discriminator density $d_D = d_G$
 2: **for** $t$ in $[1, \cdots, T]$ **do**
 3:    **for** $n$ in $[1, \cdots, N]$ **do**
 4:       Compute the loss of discriminator $\mathcal{L}_D(\boldsymbol{\theta}_D)$
 5:       $\mathcal{L}_D(\boldsymbol{\theta}_D).backward()$
 6:    **end for**
 7:    **if** $t \mod \Delta T_D == 0$ **then**
 8:       Compute the loss of generator $\mathcal{L}_G(\boldsymbol{\theta}_G)$
 9:       $\mathcal{L}_G(\boldsymbol{\theta}_G).backward()$
10:       Compute time-averaged BR with Equation 3
11:       **if** BR $> B_+$ and $d_D < d_{\max}$ **then**
12:          Increase the density of discriminator from $d_D$ to $\min(d_{\max}, d_D + \Delta d)$.
13:       **else if** BR $> B_+$ and $d_D == d_{\max}$ **then**
14:          Apply DST to $D$
15:       **else if** BR $< B_-$ **then**
16:          Decrease the density of discriminator from $d_D$ to $\max(0\%, d_D - \Delta d)$.
17:       **end if**
18:    **end if**
19:    **if** $t \mod \Delta T_G == 0$ **then**
20:       Apply DST to $G$
21:    **end if**
22: **end for**

---

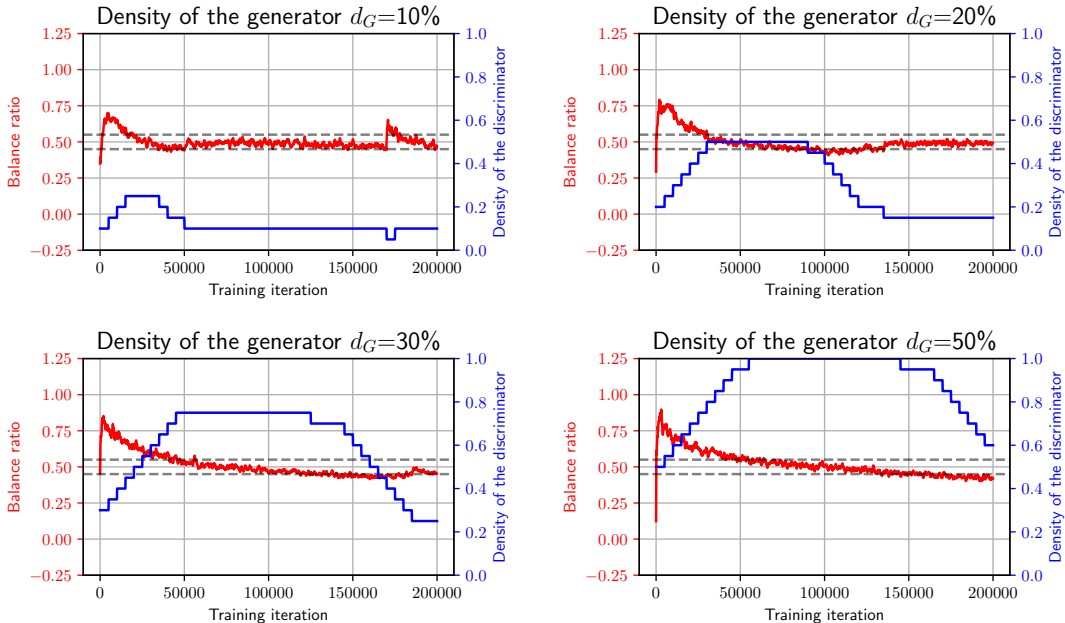

Figure 5: Balance ratio and discriminator density evolution during training for $\text{ADAPT}_{\text{relax}}$ on BigGAN (CIFAR-10). Dashed lines represent BR values of 0.45 and 0.55.

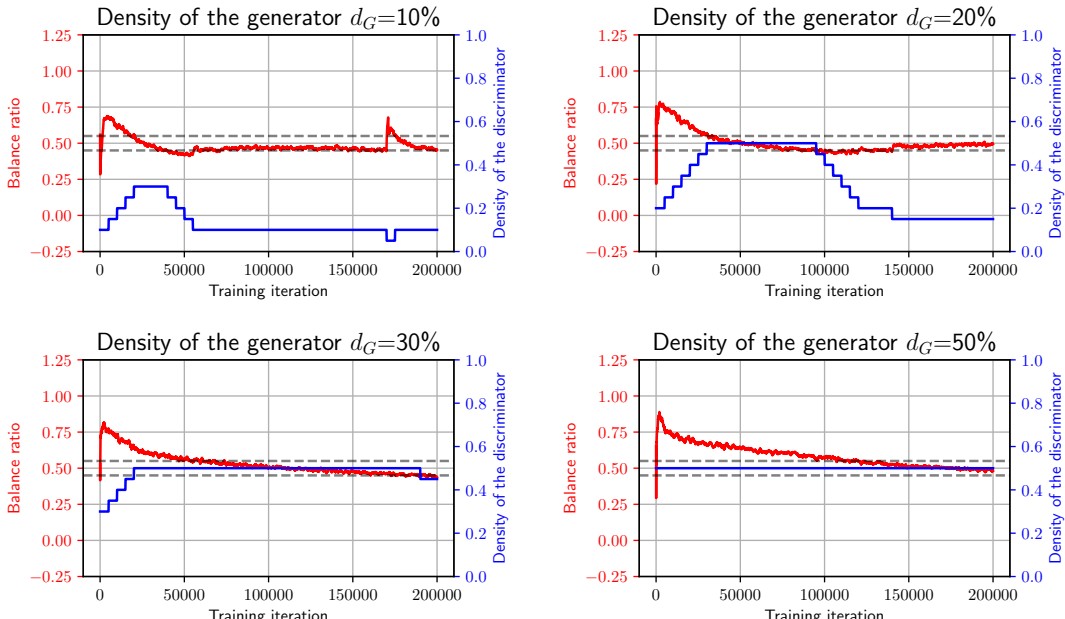

Figure 6: Balance ratio and discriminator density evolution during training for $\text{ADAPT}_{\text{strict}}$ on BigGAN (CIFAR-10). Dashed lines represent BR values of 0.45 and 0.55.

# D  `ADAPT` balance ratio evolution

In this section, we show that `ADAPT` methods are able to maintain a BR throughout training. We show the time evolution of BR and discriminator density for BigGAN on the CIFAR-10 dataset.

Results of $\text{ADAPT}_{\text{relax}}$ and $\text{ADAPT}_{\text{strict}}$ are shown in Figure 5 and Figure 6. It clearly illustrates the ability of `ADAPT` to keep the BR controlled during GAN training.

Table 6: FID ($\downarrow$) of different sparse training methods along with post-hoc pruning baseline **with no constraint on the density of the discriminator**. Best results are in **bold**; second-best results are underlined.

| Dataset | CIFAR-10 (SNGAN) | | | | STL-10 (SNGAN) | | | | CIFAR-10 (BigGAN) | | | |
|---|---|---|---|---|---|---|---|---|---|---|---|---|
| Generator density | 10% | 20% | 30% | 50% | 10% | 20% | 30% | 50% | 10% | 20% | 30% | 50% |
| (Dense Baseline) | 10.74 | | | | 29.71 | | | | 8.11 | | | |
| Post-hoc pruning | 20.89 | 14.07 | 12.99 | 11.90 | 57.28 | 37.12 | 31.98 | **29.70** | 15.44 | 10.84 | 9.65 | 8.77 |
| STATIC-Balance | 26.75 | 19.04 | 15.05 | 12.24 | 48.18 | 44.67 | 41.73 | 37.68 | 16.98 | 12.81 | 10.33 | 8.47 |
| STATIC-Strong | 26.79 | 19.65 | 14.38 | 11.91 | 52.48 | 43.85 | 42.06 | 37.47 | 23.48 | 14.26 | 11.19 | 8.64 |
| ◆ SDST-Balance-SET | 26.23 | 17.79 | 13.21 | 11.79 | 56.41 | 46.58 | 39.93 | 30.37 | 12.41 | 9.87 | 9.13 | **8.01** |
| ◆ SDST-Strong-SET | 16.49 | 13.36 | **11.68** | 10.68 | 67.37 | 49.96 | 37.99 | 31.08 | 18.94 | 9.64 | 8.75 | 8.36 |
| ● SDST-Balance-RigL | 27.06 | 16.36 | 14.00 | 12.28 | 43.08 | 33.90 | 31.83 | 30.30 | 12.45 | 9.42 | 8.86 | 8.03 |
| ● SDST-Strong-RigL | 17.02 | 13.86 | 12.51 | 11.35 | 53.65 | 33.25 | **31.41** | 30.18 | 10.58 | 9.11 | 8.69 | 8.33 |
| ADAPT$_{\text{relax}}$ (Ours) | **14.19** | **13.19** | 12.38 | **10.60** | **35.98** | **33.06** | 31.71 | 29.96 | **10.19** | **8.56** | **8.36** | 8.22 |

# E  More experiment results

## E.1  IS and FID for the CIFAR-10 dataset

In this section, we present corresponding IS scores results for Table 1 and Table 2. The results are shown in Table 8 and Table 9, respectively. We also include FID results of CIFAR-10 test set in Table 10.

## E.2  Naively applying DST to both the generator and the discriminator

In this section, we follow STU-GAN to compare the baseline where applying DST on both generators and discriminators. We name it `DST-bothGD`.

We test on SNGAN (CIFAR-10) with $\Delta T_D = 1000$, $\Delta T_G = 500$, and $\gamma = 0.5$. Note that we use the balance strategy where $d_G = d_D$. The reason is that the strong strategy uses a dense discriminator, and it does not make sense to apply DST to a dense network.

We show the results in Table 7. It shows that it generates unstable results and consistently performs worse than `SDST-Strong`. So we do not compare such baseline in the main body of the paper.

## E.3  Post-hoc pruning baseline

In this section, we compare different sparse training methods with post-hoc magnitude pruning [68] baseline. Magnitude pruning involves first training a dense generator, then pruning its weights globally based on their magnitudes. The pruned generator is then fine-tuned with the dense discriminator. We perform additional fine-tuning for $50\%$ of the original total iterations. Results are presented in Table 6.

Our experimental results clearly demonstrate the advantages of dynamic sparse training over post-hoc magnitude pruning. The latter typically requires around 150% normalized training FLOPs, while DST methods constantly achieve comparable or better performance with significantly reduced computational cost.

# F  A detailed comparison of training costs

In this section, we include the detailed computational cost of all sparse training methods. More specifically, we take into account the density redistribution over different layers in this section. Also, we make an assumption that the computational overhead introduced by computing BR can be neglected.[4]

Here we provide training costs for the **strict** setting in Table 12.

---

[4]This is true if we compute BR less frequently.

Table 7: FID ($\downarrow$) of different sparse training methods on CIFAR-10 datasets with no constraint on the density of the discriminator. Best results are in **bold**; second-best results are underlined.

| Dataset | CIFAR-10 | | | |
|---|---|---|---|---|
| Generator density | 10% | 20 % | 30 % | 50 % |
| (Dense Baseline) | 10.74 | | | |
| Static-Balance | 26.75 | 19.04 | 15.05 | 12.24 |
| Static-Strong | 26.79 | 19.65 | 14.38 | 11.91 |
| ◆ DST-bothGD-SET | 20.57 | 14.90 | 12.58 | 11.86 |
| ● DST-bothGD-RigL | 31.95 | 17.99 | 13.24 | 12.47 |
| ◆ SDST-Balance-SET | 26.23 | 17.79 | 13.21 | 11.79 |
| ◆ SDST-Strong-SET | 16.49 | 13.36 | **11.68** | 10.68 |
| ● SDST-Balance-RigL | 27.06 | 16.36 | 14.00 | 12.28 |
| ● SDST-Strong-RigL | 17.02 | 13.86 | 12.51 | 11.35 |
| ADAPT$_{relax}$ (Ours) | **14.19** | **13.19** | 12.38 | **10.60** |

Table 8: IS (higher is better) of different sparse training methods. There is no constraint on the density of the discriminator, i.e., $d_{max} = 100\%$.

| Dataset | SNGAN(CIFAR-10) | | | | SNGAN(STL-10) | | | | BigGAN(CIFAR-10) | | | | BigGAN(TinyImageNet) | | | |
|---|---|---|---|---|---|---|---|---|---|---|---|---|---|---|---|---|
| Generator density | 10% | 20 % | 30 % | 50 % | 10% | 20 % | 30 % | 50 % | 10% | 20 % | 30 % | 50 % | 10% | 20 % | 30 % | 50 % |
| (Dense Baseline) | 8.48 | | | | 9.16 | | | | 8.99 | | | | 14.65 | | | |
| Static-Balance | 7.24 | 7.83 | 8.06 | 8.38 | 7.94 | 8.19 | 8.44 | 8.69 | 7.99 | 8.24 | 8.68 | 8.90 | 10.65 | 12.28 | 13.41 | 13.57 |
| Static-Strong | 7.52 | 8.03 | 8.32 | 8.45 | 7.70 | 8.22 | 8.35 | 8.70 | 7.75 | 8.13 | 8.52 | 8.99 | 10.45 | 12.56 | 13.61 | 13.73 |
| ◆ SDST-Balance-SET | 7.28 | 7.89 | 8.22 | 8.38 | 8.43 | 8.92 | 9.26 | 9.31 | 8.62 | 8.67 | 8.82 | 8.98 | 11.75 | 12.60 | 12.30 | 12.21 |
| ◆ SDST-Strong-SET | 8.37 | 8.54 | 8.57 | 8.60 | 7.65 | 8.53 | 9.39 | 9.21 | 8.16 | 8.78 | 8.85 | 9.06 | 12.75 | 12.84 | 12.46 | 13.73 |
| ● SDST-Balance-RigL | 7.19 | 7.94 | 8.18 | 8.34 | 8.98 | 9.07 | 9.12 | 9.28 | 8.64 | 8.71 | 8.91 | 8.93 | 12.67 | 13.32 | 13.18 | 13.61 |
| ● SDST-Strong-RigL | 8.32 | 8.52 | 8.59 | 8.57 | 8.15 | 9.10 | 9.16 | 9.17 | 8.65 | 8.72 | 8.97 | 9.00 | 13.32 | 13.35 | 13.60 | 14.47 |
| ADAPT$_{relax}$ (Ours) | 8.42 | 8.44 | 8.54 | 8.60 | 9.08 | 9.29 | 9.06 | 9.26 | 8.74 | 9.07 | 8.98 | 9.00 | 13.09 | 13.57 | 13.68 | 15.77 |

Table 9: IS (higher is better) of different sparse training methods. The density of the discriminator is constrained to be lower than $d_{max} = 50\%$.

| Dataset | SNGAN(CIFAR-10) | | | | SNGAN(STL-10) | | | | BigGAN(CIFAR-10) | | | | BigGAN(TinyImageNet) | | | |
|---|---|---|---|---|---|---|---|---|---|---|---|---|---|---|---|---|
| Generator density | 10% | 20 % | 30 % | 50 % | 10% | 20 % | 30 % | 50 % | 10% | 20 % | 30 % | 50 % | 10% | 20 % | 30 % | 50 % |
| (Dense Baseline) | 8.48 | | | | 9.16 | | | | 8.99 | | | | 14.65 | | | |
| Static-Balance | 7.24 | 7.83 | 8.06 | 8.38 | 7.94 | 8.19 | 8.44 | 8.69 | 7.99 | 8.24 | 8.68 | 8.90 | 10.65 | 12.28 | 13.41 | 13.57 |
| Static-Strong | 7.85 | 8.14 | 8.31 | 8.38 | 7.89 | 8.22 | 8.38 | 8.69 | 7.75 | 8.03 | 8.52 | 8.90 | 9.99 | 11.61 | 13.77 | 13.57 |
| ◆ SDST-Balance-SET | 7.28 | 7.89 | 8.22 | 8.38 | 8.43 | 8.92 | 9.26 | 9.31 | 8.62 | 8.67 | 8.82 | 8.98 | 11.75 | 12.60 | 12.30 | 12.21 |
| ◆ SDST-Strong-SET | 8.33 | 8.53 | 8.40 | 8.38 | 8.50 | 8.77 | 9.46 | 9.26 | 8.55 | 8.77 | 8.84 | 8.98 | 12.00 | 12.87 | 12.16 | 12.21 |
| ● SDST-Balance-RigL | 7.19 | 7.94 | 8.18 | 8.34 | 8.98 | 9.07 | 9.12 | 9.28 | 8.64 | 8.71 | 8.91 | 8.93 | 12.67 | 13.32 | 13.18 | 13.61 |
| ● SDST-Strong-RigL | 8.24 | 8.48 | 8.37 | 8.34 | 8.28 | 9.05 | 9.11 | 9.28 | 8.61 | 8.83 | 8.84 | 8.93 | 12.04 | 12.66 | 13.57 | 13.61 |
| ADAPT$_{strict}$ (Ours) | 8.27 | 8.36 | 8.48 | 8.47 | 8.98 | 9.17 | 9.20 | 9.19 | 8.90 | 8.89 | 8.92 | 9.10 | 13.85 | 13.61 | 14.05 | 14.40 |

Table 10: FID of test set ($\downarrow$) of different sparse training methods on SNGAN (CIFAR-10) dataset. Best results are in **bold**; second-best results are underlined.

| Maximal discriminator density $d_{max}$ | 100 % | | | | 50 % | | | |
|---|---|---|---|---|---|---|---|---|
| Generator density | 10% | 20 % | 30 % | 50 % | 10% | 20 % | 30 % | 50 % |
| (Dense Baseline) | 13.32 | | | | | | | |
| Static-Balance | 29.56 | 21.79 | 17.80 | 14.94 | 29.56 | 21.79 | 17.80 | 14.94 |
| Static-Strong | 29.50 | 22.45 | 17.12 | 14.58 | 24.62 | 19.43 | 16.32 | 14.94 |
| ◆ SDST-Balance-SET | 28.84 | 20.31 | 15.95 | 14.35 | 28.84 | 20.31 | 15.95 | **14.35** |
| ◆ SDST-Strong-SET | 19.16 | 16.12 | **14.45** | 13.50 | 18.38 | **15.33** | **14.78** | **14.35** |
| ● SDST-Balance-RigL | 29.77 | 19.02 | 16.68 | 15.05 | 29.77 | 19.02 | 16.68 | 15.05 |
| ● SDST-Strong-RigL | 19.72 | 16.50 | 15.20 | 14.09 | 17.92 | 15.51 | 15.52 | 15.05 |
| ADAPT$_{relax}$ (Ours) | **16.82** | **15.85** | 15.14 | **13.37** | - | - | - | - |
| ADAPT$_{strict}$ (Ours) | - | - | - | - | **17.19** | 15.57 | 14.92 | 14.80 |

Table 11: FID of test set ($\downarrow$) of different sparse training methods on BigGAN (CIFAR-10) dataset. Best results are in **bold**; second-best results are underlined.

| Maximal discriminator density $d_{\max}$ | **100 %** | | | | **50 %** | | | |
|---|---|---|---|---|---|---|---|---|
| Generator density | 10% | 20 % | 30 % | 50 % | 10% | 20 % | 30 % | 50 % |
| (Dense Baseline) | | | | 10.36 | | | | |
| Static-Balance | 19.58 | 15.63 | 13.21 | 10.92 | 19.58 | 15.63 | 13.21 | 10.92 |
| Static-Strong | 26.08 | 15.82 | 13.47 | 10.95 | 22.04 | 16.39 | 13.73 | 10.92 |
| ◆ SDST-Balance-SET | 14.90 | 12.77 | 11.82 | **10.68** | 14.90 | 12.77 | 11.82 | 10.68 |
| ◆ SDST-Strong-SET | 21.63 | 11.92 | 11.27 | 10.75 | 14.53 | 11.83 | 10.96 | 10.68 |
| ● SDST-Balance-RigL | 14.86 | 12.03 | 11.30 | **10.68** | 14.86 | 12.03 | 11.30 | 10.68 |
| ● SDST-Strong-RigL | 13.35 | 11.58 | 11.00 | 10.88 | 12.59 | 12.03 | **10.89** | 10.68 |
| ADAPT$_{\text{relax}}$ (Ours) | **12.71** | **11.02** | **10.62** | 10.80 | - | - | - | - |
| ADAPT$_{\text{strict}}$ (Ours) | - | - | - | - | **11.83** | **11.22** | 10.92 | **10.33** |

Table 12: Normalized training FLOPs ($\downarrow$) of different sparse training methods. **The density of the discriminator is constrained to be lower than 50%.**

| Dataset | CIFAR-10 (SNGAN) | | | | STL-10 (SNGAN) | | | | CIFAR-10 (BigGAN) | | | | TinyImageNet (BigGAN) | | | |
|---|---|---|---|---|---|---|---|---|---|---|---|---|---|---|---|---|
| Generator density | 10% | 20% | 30% | 50% | 10% | 20% | 30% | 50% | 10% | 20% | 30% | 50% | 10% | 20% | 30% | 50% |
| (Dense Baseline) | 100% ($2.67 \times 10^{17}$) | | | | 100% ($3.94 \times 10^{17}$) | | | | 100% ($6.81 \times 10^{17}$) | | | | 100% ($9.85 \times 10^{17}$) | | | |
| Static-Balance | 8.97% | 17.08% | 26.25% | 47.25% | 27.30% | 47.14% | 59.22% | 73.35% | 9.79% | 19.02% | 28.66% | 49.03% | 23.25% | 44.87% | 60.91% | 79.29% |
| Static-Strong | 30.89% | 33.58% | 37.17% | 47.25% | 70.65% | 71.48% | 72.14% | 73.35% | 42.66% | 43.69% | 45.10% | 49.03% | 41.52% | 55.03% | 66.29% | 79.29% |
| ◆ SDST-Balance-SET | 9.78% | 18.91% | 28.35% | 48.44% | 27.55% | 47.60% | 60.17% | 75.38% | 10.35% | 20.12% | 29.96% | 49.82% | 21.13% | 37.06% | 48.83% | 65.58% |
| ◆ SDST-Strong-SET | 31.87% | 35.51% | 39.53% | 48.44% | 70.95% | 71.97% | 73.07% | 75.38% | 43.25% | 44.80% | 46.42% | 49.82% | 39.28% | 47.31% | 54.11% | 65.58% |
| ● SDST-Balance-RigL | 10.71% | 17.43% | 25.66% | 43.56% | 29.51% | 50.41% | 63.34% | 79.03% | 9.92% | 19.30% | 28.90% | 48.31% | 24.97% | 43.86% | 57.26% | 76.75% |
| ● SDST-Strong-RigL | 31.22% | 33.93% | 36.63% | 43.56% | 72.95% | 75.05% | 76.42% | 79.03% | 42.80% | 44.08% | 45.37% | 48.31% | 43.76% | 53.71% | 63.05% | 76.75% |
| ADAPT$_{\text{strict}}$ (Ours) | 24.23% | 27.55% | 31.70% | 37.83% | 50.91% | 70.18% | 75.99% | 80.68% | 10.32% | 23.69% | 31.54% | 33.83% | 34.42% | 51.68% | 62.34% | 77.46% |

