# OpenReview forum: "Balanced Training for Sparse GANs"
_NeurIPS.cc/2023/Conference — NeurIPS 2023 poster_

### Official Review · Reviewer_sFon · 2023-07-04

**Soundness:** 2 fair
**Presentation:** 3 good
**Contribution:** 4 excellent
**Rating:** 5
**Confidence:** 3

**Summary:**

This work proposes a metric named balance ratio to represent the balance between the generator and discriminator in dynamic sparse training, and furthermore proposes balanced dynamic sparse training to balance the performance and computation cost.

**Strengths:**

(++) There are not many previous works that tried to apply DST to GANs, so this research is very valuable, in my opinion. The insight of the problem of balance between the generator and discriminator is also accurate.
(++) The motivation is clearly explained and discussed well.

**Weaknesses:**

(----) The experiments are the main problem. The baselines (SNGAN, BigGAN) seem to be out of date. The involved datasets are all small, but larger images (e.g., FFHQ) are not utilized. Recently, GAN frameworks can even generate images larger than 1024x1024. Generating small images is not so challenging today, as powerful frameworks, e.g., diffusion models, have achieved great success.  In my opinion, at least 256x256 images should be considered to show the value of the proposed method.

**Questions:**

Could you please give a reasonable explanation for only involving small datasets? If there is a reason I did not notice, I will correspondingly update my score.

**Limitations:**

I did not find the discussion of limitations. I also did not notice the potential limitations of this work.

---

> ### Author Rebuttal · Authors · 2023-08-09
>
> We sincerely appreciate the reviewer's recognition of the value and motivation behind our work. We would like to address the reviewer's concerns as follows:
>
> > **Q1. The experiments are the main problem. The baselines (SNGAN, BigGAN) seem to be out of date.**
>
> Thank you for your valuable input. However, we respectfully disagree with the notion that BigGAN is outdated compared to SOTA, e.g. StyleGAN-XL. Although StyleGAN is currently the SOTA model for generating high-resolution images, it has some limitations that make it less versatile than BigGAN. For instance, research [8] has shown that the StyleGAN family struggles with generating images that have high inter-class variations, such as those in ImageNet. Therefore, we believe it is reasonable to conduct our experiments on BigGAN, which is still a highly effective and versatile model for image generation tasks. However, we appreciate your suggestion and will keep the latest SOTA models in mind for future work.
>
> > **Q2. In my opinion, at least 256x256 images should be considered to show the value of the proposed method.**
>
> We want to explain that **(1) we mainly design our experiments following previous baselines, and (2) our contribution is not solely ADAPT's performance.**
>
> 1. It is worth noting that the field of GAN dynamic sparse training (DST) is relatively new, and the pioneering work, STU-GAN, mainly draws its conclusions based on the CIFAR-10 dataset. In our experiments, we have observed excellent performance of our proposed methods on CIFAR-10, STL-10, and TinyImageNet datasets. Additionally, running experiments on larger datasets and higher resolutions can be computationally demanding, requiring multiple GPUs and several days to complete a single case. Given these resource constraints, conducting extensive experiments involving multiple density ratios, SDST variants, and two settings on larger datasets may currently be computationally heavy.
>
> 2. The primary goal of our paper extends beyond the state-of-the-art (SOTA) method.
> To provide a concise summary, we accomplish the following:
> (1) Propose a novel metric to study the balance in sparse GAN training.
> (2) Introduce and analyze the behaviors of various strategies for STU-GAN, offering valuable insights into the effectiveness and limitations of it.
> (3) Identify and propose solutions for certain limitations of STU-GAN, thereby contributing to the advancement of dynamic sparse training in the GAN domain.
> We firmly believe that these findings not only enhance the performance of STU-GAN but also have the potential to pave the way for further research and advancements in this field.
>
> Nonetheless, we highly value the reviewer's suggestion, and we are actively working on testing our methods on larger datasets. As soon as these experiments are completed, we will incorporate the results into the next version of our work, providing a comprehensive evaluation of our proposed methods.

---

> > ### Comment · Reviewer_sFon · 2023-08-15
> >
> > I appreciate the author's efforts in their rebuttal. The additional experiment results have addressed most of my concerns.
> >
> > I still uphold that BigGAN is outdated, at least for CIFAR-10, because the mentioned StyleGAN-XL has achieved an FID of 1.85 on CIFAR-10, which will be very difficult to be surpassed for BigGAN.  As the research interest of this work is not lightweight design, there is also no reason to consider saving computing resources especially.
> >
> > However, I agree that the goal of the paper is not to achieve SOTA performance. For most SOTA frameworks, spectral normalization (SN) will be employed to stabilize training. The authors have conducted experiments on SNGAN, demonstrating that the method can work well with the SN technique. Note that SN is also a technique for suppressing mode collapse. As a result, the proposed method is very likely to be effective for SOTA frameworks. It would be better if the authors could provide a simple experiment, applying the proposed method on a SOTA framework with one large dataset, but there is no need to compare it to other SOTA models.
> >
> > Meanwhile, it is not a well-studied topic of DST in GANs. GANs also need new ideas and improvements to gain more advantages in competition with other frameworks. This paper also provides a good motivation, technical analysis, and method. Thus, compared to the lack of an experiment, I think the advantage of this work outweighs the disadvantages, so I have raised my score.

---

> > > ### Author Response · Authors · 2023-08-15
> > > **Thank you for your comment**
> > >
> > > We are grateful for the time and efforts the reviewer spent in reviewing and providing valuable comments and thank you for considering raising your rating. Your suggestions for the experiments are greatly appreciated, and we will certainly incorporate the results into the manuscript once they are available.

---

### Official Review · Reviewer_PnSi · 2023-07-04

**Soundness:** 3 good
**Presentation:** 3 good
**Contribution:** 2 fair
**Rating:** 4
**Confidence:** 5

**Summary:**

This paper presents a method for dynamic sparse training for GANs. In particular, the authors propose the balance ratio to study the balance status between the generator and discriminator. In addition, a balanced dynamic sparse training strategy is designed by applying BR to achieve a good trade-off between performance and computational cost. Experimental results proved the effectiveness of the proposed method.

**Strengths:**

(1) The motivation of balancing sparse GAN training under resource budget is well presented.

(2) The explanations and illustrations of the balance ratio and balanced dynamic sparse training are well-formulated and mostly clear and intuitive.

**Weaknesses:**

(1) The motivation of applying DST to GAN is not that novel to the community and the authors listed STU-GAN as an example. In addition, the experimental improvements over STU-GAN is trivial according to Table 1-3.

(2) The normalized training FLOPs in Table 3 fluctuate a lot and are not linearly consistent with the generator density. It contradicts the basic belief of balanced model size between the generator and discriminator for stable GAN training.

(3) The DST strategy is in-time over-parameterization, what is the difference to a dynamic-ratio Dropout?

(4) When the DST strategy is applied, the balance ratio cannot explain away other important factors for the GAN performance. For example, when spectral normalization is used in the discriminator, the GAN training can be very stable among a lot of practice even for unbalanced model size between the generator and discriminator.

**Questions:**

No.

**Limitations:**

No.

---

> ### Author Rebuttal · Authors · 2023-08-09
>
>
> We thank the reviewer for recognizing our work is well-motivated, well-formulated and clear. We hereby address the reviewer's questions.
>
> > **Q1. The motivation of applying DST to GAN is not that novel.**
>
> **We want to point out that though STU-GAN [6] is the first to apply DST on GAN, it has its limitations and it requires a better understanding.** Our work is novel as it provides a way to analyze DST for GANs, identifies STU-GAN's limitations, and proposes improvements to address these shortcomings.
>
> More specifically, in our Section 5, we study STU-GAN through BR and highlight several limitations of it:
>
> 1. STU-GAN requires a pre-defined density for the discriminator, lacking a principled method for choosing this value, while the chosen density can significantly impact the final performance of STU-GAN (see Figure 1).
>
> 2. STU-GAN may fail when the discriminator is initialized to be weak.
>
> In essence, STU-GAN only mitigates the unbalance problem when the discriminator is stronger than the generator. The limitation originates from the fact that (1) STU-GAN does not have a principled way to measure the unbalance, (2) STU-GAN only adjusts the generator.
>
> Our work identifies these shortcomings through BR and proposes to address the limitations of STU-GAN.
>
> > **Q2. In addition, the experimental improvements over STU-GAN are trivial according to Table 1-3.**
>
> We appreciate the reviewer's input, but we respectively do not agree with the reviewer's opinion as (1) the improvement is not trivial, and (2) our goal is not solely to become the SOTA method.
>
> 1. **Our improvement is not trivial.** Our improvement is most prominent in very sparse cases. For example, consider the second-best method SDST-Strong-RigL which is well-performing and stable:
> (1) Table 1, 10\% gen density, SNGAN on CIFAR-10. Our method outperforms it by 2.83 FID with 63\% FLOPs.
> (2) Table 1, 10\% gen density, SNGAN on STL-10. Our method outperforms it by 17.67 FID with 53\% FLOPs.
> (3) Table 1, 10\% gen density, BigGAN on Tiny-ImageNet. Our method outperforms it by 1.72 FID with 59\% FLOPs.
> Similar favorable outcomes are observed in Table 2. Therefore, our method improves the performance of STU-GAN while significantly reducing computational costs.
>
> 2. **The primary goal of our paper extends beyond the SOTA method.** To summarize, (1) we propose a metric to study the balance in sparse GAN training, (2) we introduce different strategies for STU-GAN and study the behaviors of them, (3) we identify and propose solutions for certain limitations of STU-GAN. We believe these findings may pave the way for further research in this domain.
>
> > **Q3. The normalized training FLOPs in Table 3 fluctuate a lot and are not linearly consistent with the generator density. It contradicts the basic belief of balanced model size between the generator and discriminator for stable GAN training.**
>
> We appreciate the reviewer for bringing up this question, and we would like to provide some insights into why the normalized training FLOPs (sum of the generator and discriminator) do not exhibit "linear consistent (scaling)"  with the generator density:
>
> 1. **The discriminator density is not fixed.** In our experiments, we observed that the discriminator's density may first increase and then decrease until it stabilizes during the training process. This behavior can be observed in Figure 5 and Figure 6, which helps explain why the total training FLOPs do not linearly scale with the generator density.
>
> 2. **The balancing density of the discriminator may not linearly scale with the generator density.** It is important to note that the generator's capacity does not necessarily linearly scale with its density. For example, having a generator with twice the number of parameters does **NOT** imply a proportional increase in representation power or capacity. The same holds true for the discriminator.  As a result, the non-linear relationship between generator density and discriminator density further contributes to the non-linear scaling of total training FLOPs.
>
> 3. **Moreover, due to DST/sparse initialization, the allocation of parameters across different layers may vary for different G/D densities and even different epochs.**
>
> Hence, we do not expect the total training FLOPs to exhibit linear scaling with the generator density.
>
> > **Q4. The DST strategy is in-time over-parameterization, what is the difference to a dynamic-ratio Dropout?**
>
> We appreciate the reviewer's question. The key distinctions between ITOP (or DST in general) and dynamic-ratio Dropout lie in **resulting network sparsity**.
>
> During training, pruned weights in DST are set to zero and do not receive updates. In contrast, while dropout sets weights' gradients to zero, their magnitudes remain nonzero and may still receive updates when momentum is used. Consequently, DST results in a sparse network with many zero weights after training, while dynamic-ratio Dropout produces a (normal) dense network.
>
> We hope this clarifies the distinction between ITOP and dynamic-ratio Dropout.
>
> > **Q5. When the DST is applied, BR cannot explain away other important factors for the GAN performance. For example, when spectral normalization (SN) is used in the discriminator, the GAN training can be very stable among a lot of practice even for unbalanced model size between the generator and discriminator.**
>
> We thank the reviewer for the question. However, we want to emphasize that **(1) SN is indeed used in our models, and (2) SN alone is not enough to enable balanced training for sparse GANs.**
>
> We want to kindly point out that in our experiments (Section 5), we indeed applied SN to the discriminator. The results shows:
> 1. SN alone is not sufficient to stabilize the unbalanced model size between the two components. Despite using SN, the sparse GAN training may still exhibit instability due to the unbalanced components.
> 2. The BR is able to quantify the unbalance of the two components when SN is applied.

---

> > ### Author Response · Authors · 2023-08-21
> >
> > Dear Reviewer PnSi,
> >
> > Following our recent rebuttal submission, we wanted to ensure you've had an opportunity to review our responses. As the reviewer-author discussion period is drawing to a close, we'd value any feedback you might have.
> >
> > Thank you for your attention and understanding.
> >
> > Best regards,
> >
> > Authors

---

### Official Review · Reviewer_6W4E · 2023-07-09

**Soundness:** 4 excellent
**Presentation:** 3 good
**Contribution:** 4 excellent
**Rating:** 8
**Confidence:** 4

**Summary:**

The paper addresses the challenge of reducing the computational complexity of training GANs by leveraging dynamic sparse training (DST) techniques. The authors propose a novel metric called the balance ratio (BR) to quantify the balance between the sparse generator and discriminator during GAN training. They also introduce a method called balanced dynamic sparse training (ADAPT) to control the BR and achieve a balance between performance and computational cost.

The paper begins by providing a thorough background on GANs, sparse training, and the challenges associated with applying DST to GANs. The motivation for the research is clearly explained, emphasizing the need for efficient training methods without sacrificing performance. The proposed metric, BR, is introduced and its significance in measuring the balance between the generator and discriminator is well-established.

The methodology section is comprehensive, detailing the steps involved in ADAPT. The authors describe the specific modifications to the GAN training process, incorporating sparse training techniques and controlling the BR. The mathematical formulations and algorithms are clearly presented, making it easier to understand the implementation details.

The paper is generally well-written and structured, with a logical flow of ideas. However, there are a few areas where the clarity could be improved. In some sections, the technical details and explanations are a bit dense, making it challenging for readers unfamiliar with the topic to follow along. Providing more intuitive explanations or examples could enhance the accessibility of the paper.

**Strengths:**

1. The paper is well-motivated.
2. Using balance-ratio as a metric to understand generator and discriminator sparsity and affects is interesting and seems very useful.
3. The performance improvement over STATIC demonstrate the effectiveness of DDST.

**Weaknesses:**

The technical details and explanations are a bit dense. Providing more intuitive explanations or examples could enhance the accessibility of the paper. Not required but this can be part of Appendix.

**Questions:**



**Limitations:**

---

> ### Author Rebuttal · Authors · 2023-08-09
>
> We sincerely appreciate the reviewer's insightful feedback, which recognizes our work is well-motivated, interesting, and effective. Your positive evaluation is both encouraging and valuable.
>
> In response to the reviewer's valuable suggestion, we will follow the reviewer's suggestion to provide additional details and establish stronger connections between different sections of the paper in the subsequent versions, thereby improving the overall clarity and coherence of our work.
>
> If there are any further questions or concerns, please raise them and we are more than willing to answer them.

---

> > ### Comment · Reviewer_6W4E · 2023-08-21
> >
> > I have read the rebuttals and I am satisfied.

---

### Official Review · Reviewer_cAMN · 2023-07-15

**Soundness:** 3 good
**Presentation:** 3 good
**Contribution:** 3 good
**Rating:** 5
**Confidence:** 4

**Summary:**

Motivated by the identified imbalance between the generator and discriminator during sparse GAN training, this work proposes a quantitative metric dubbed balance ratio as an indicator for the degree of balance in sparse GAN training. Leveraging this metric, this work further proposes the ADAPT framework to dynamically adjust the sparsity of the discriminator towards balanced GAN training. Experiments across various datasets validate the effectiveness of the proposed method in achieving a good trade-off between performance and computational cost.

**Strengths:**

1. This work is well-written with a clear logical flow. The well-organized structure from observations to understandings and solutions is appreciated.

2. The proposed balance ratio can well indicate the balance between the generator and discriminator during GAN training, which can potentially serve as a useful metric for the community.

**Weaknesses:**

1. The imbalance between the generator and discriminator has been extensively studied and it is not clear why the proposed quantitative metric (i.e., the balance ratio) can outperform previous indicators or solutions under the scenario of sparse GAN training. The authors are expected to provide a literature review and discuss the key advantage of the proposed method that makes it particularly suitable for sparse GAN training.

2. It is not clear what is the rationale for proposing the two variants ADAPT_relax and ADAPT_strict. What is the expectation of the performance ranking between the two according to the claim "a more interesting observation is that ADAPT_strict sometimes outperforms ADAPT_relax"?

3.  For the experimental results, only limited/insufficient baselines are considered. The authors are expected to benchmark the proposed framework with other sparse GAN training methods, e.g., [1][2] cited below.

[1] "Data-efficient gan training beyond (just) augmentations: A lottery ticket perspective", T. Chen et al., NeurIPS'21.

[2] "Don't be so dense: sparse-to-sparse gan training without sacrificing performance", T. Chen et al., IJCV'23.

4. Since only unstructured sparsity is considered, the reported FLOPs reduction cannot be turned into real-device speedup. The authors are expected to perform experiments under structured sparsity to validate whether the proposed method is still effective.

**Questions:**

I have listed my questions in the weakness section. I am willing to adjust my scores if my concerns are properly addressed.

Minor: Line 256 "while the density of the generator is dynamically adjusted with DDA" => Here "generator" should be the "discriminator"?

**Limitations:**

This work targets improving the training efficiency of GANs, thus not suffering from obvious negative societal impact.

---

> ### Author Rebuttal · Authors · 2023-08-09
>
> We thank the reviewer for acknowledging that our work is well-written and the proposed metric is useful. We hereby address the reviewer's concerns:
>
> > **Q1. It is not clear why the proposed metric can outperform previous indicators or solutions. The authors are expected to provide a literature review and discuss the key advantage of the proposed method that makes it particularly suitable for sparse GAN training.**
>
> We express our gratitude to the reviewer for providing constructive suggestions. In fact, in our paper, we have included several related papers [1,2,5,6,7]. We hereby provide a more detailed literature review in the **global rebuttal** and we intend to include it in our forthcoming versions as well.
>
> > **Q2. It is not clear what is the rationale for proposing the two variants of ADAPT.**
>
> We deeply appreciate the reviewer's insightful question. If we understand correctly, the reviewer raises the question regarding the "rationale" of the more difficult setting and its corresponding variant $ADAPT_{strict}$ (Please correct us if we are wrong). We want to clarify that we do so to **(1) provide a more comprehensive study by evaluating our proposed approach in a different setting, and (2) to further push the limit of DST to achieve even greater reductions in computational costs.**
>
> As the reviewer mentions, normally, we would only consider the setting where we can choose  discriminators with arbitrary densities (as considered in STU-GAN [6]), i.e., the relaxed setting. Other than that, we further introduce the strict setting, which is much more difficult but ensures more computational savings, which is explained below.
>
> The fundamental distinction between the two settings lies in the density constraint imposed on the discriminator. More specifically, in the relaxed setting, the discriminator can have densities in the range $d_D \in (0\\%,100\\%]$, offering a wide range of density choices. In contrast, the strict setting with $d_D^{max}=50\\%$ limits the discriminator to choose densities in the range $d_D \in (0\\%,50\\%]$.
>
> To illustrate, let's consider a scenario where we require a stronger discriminator when the density $d_D$ is already at $50\\%$. In the relaxed setting, $ADAPT_{relax}$ has the freedom to increase the discriminator's density from $50\\%$ to $100\\%$ (resulting in a dense discriminator). However, such an option is unavailable for $ADAPT_{strict}$ since we are already utilizing the highest density permitted. Consequently, the relaxed setting ensures a minimum computational savings of approximately $50\\% * C_{G}$ where $C$ is the computational cost, whereas the strict setting guarantees at least around $50\\% * (C_{G}+C_{D})$ computational savings. In essence, the strict setting only permits the use of discriminators with low densities, but it ultimately leads to more substantial computational savings.
>
> > **Q3. What is the expectation of the results in the two settings.**
>
> Given the constraints it imposes, we expect the performance of $ADAPT_{strict}$ to be inferior compared to $ADAPT_{relax}$ as more restrictions are introduced.
>
> > **Q4. For the experimental results, only limited/insufficient baselines are considered. The authors are expected to benchmark the proposed framework with other sparse GAN training methods, e.g.,[6,7] cited below.**
>
> We express our gratitude to the reviewer for providing constructive feedback. In response, we would like to elaborate on our answers as follows:
>
> 1. We want to kindly point out that, as mentioned in our section 4 (line 137), **STU-GAN [6] is indeed included as one of our baselines**, as it is almost similar to SDST-RigL in our work.
>
> 2. The innovative work [7] primarily focuses on lottery ticket finding, which involves a costly train-prune-retrain process, aiming to improve data efficiency. However, our work centers on achieving efficient training. Therefore, we believe that [7] is not a necessary baseline for our specific research objectives.
>
> 3. As indicated in [6], STU-GAN has been shown to outperform post-hoc pruning. We have also conducted validation of this finding in Appendix E.3, Table 6, further supporting our selection of STU-GAN as a strong baseline in our study.
>
> We hope these elaborations provide better clarity on the choices we made for the baselines.
>
> > **Q5. The authors are expected to perform experiments under structured sparsity to validate whether the proposed method is still effective.**
>
> While we acknowledge the importance of structured pruning, it is crucial to take into account the current state of the GAN DST field, and the pruning field as a whole.
>
> 1. It is important to highlight that a significant number of works (almost all pruning works mentioned in Section 3) in the field continue to focus on unstructured pruning, including follow-up works of LTH, DST, foresight pruning, and others. These works have contributed invaluable insights to the pruning research community.
>
> 2. In fact, to the best of our knowledge, the only GAN DST work, i.e., STU-GAN, focuses on unstructured pruning. Our primary objective is to build upon and enhance the potential of STU-GAN by addressing its limitations and extending its capabilities. As a result, while we recognize the significance of structured pruning, it is not the foremost goal of our current work.
>
> However, we once again acknowledge that structured pruning plays a crucial role in enhancing model efficiency, and we intend to explore structured pruning extensively in our future research.
>
> > **Q6. "while the density of the generator is dynamically adjusted with DDA" => Here "generator" should be the "discriminator"**
>
> Thank you for pointing out the typo. We will fix it in the next version.

---

> > ### Comment · Reviewer_cAMN · 2023-08-16
> > **Reviewer response**
> >
> > Thank the authors for their efforts in providing the rebuttal. Most of my concerns are properly addressed. I tend to accept this paper given its current shape and will further adjust my scores based on the discussion with other reviewers.

---

> > > ### Author Response · Authors · 2023-08-19
> > >
> > > Thank you for your valuable feedback! If you have any additional questions or require further information, please feel free to raise them, and we will be more than happy to address them.

---

> > > ### Author Response · Authors · 2023-08-21
> > >
> > > Dear Reviewer cAMN,
> > >
> > > We truly appreciate your constructive feedback and the time you've taken to consider our rebuttal. We understand that you may wish to discuss with fellow reviewers. As the reviewer-author discussion period nears its end, we wish to remind you to possibly adjust our submission's score after your discussions, if you still intend to do so. Your thoughtful consideration in this matter is deeply valued.
> > >
> > > Once again, thank you for your dedication and effort throughout this review process.
> > >
> > > Best regards,
> > >
> > > Authors

---

### Author Rebuttal · Authors · 2023-08-09

We thank the reviewers for recognizing our work is well-written (cAMN, PnSi), useful (cAMN, 6W4E), effective (6W4E), well-motivated (6W4E, PnSi, sFon), and valuable (sFon). In response to the reviewers' requests, we have included the following additional content.

> **Literature review requested by reviewer cAMN**

Addressing the balance between the generator and discriminator in GAN training has been the focus of various works. However, directly applying existing methods to sparse GAN training poses challenges. For instance, [1,2] offer theoretical analyses on the issue of imbalance but may have limited practical benefits, e.g., they require training multiple generators and discriminators. Empirically, BEGAN [3] proposes to use proportional control theory to maintain a hyper-parameter $\frac{\mathbb{E}[|G(z)-D(G(z))|^\eta]}{\mathbb{E}[|x-D(x)|^\eta]}$, but it is only applicable when the discriminator is an auto-encoder. Unbalanced GAN [4] pretrains a VAE to initialize the generator, which may only address the unbalance near initialization. GCC [5] considers the balance during GAN compression, while its criterion requires a trained (dense) GAN, which is not given in the DST setting. Finally, STU-GAN [6] proposes to use DST to address the unbalance issues but may fail under certain conditions, as demonstrated in our experiments.

In summary, the existing approaches cannot be directly applied to balanced GAN DST. The only metric that could potentially be helpful for sparse GAN training is the one presented by BEGAN [3], which has restrictions on the discriminator architecture. Unlike BEGAN, our metric isn't constrained to a specific discriminator architecture. Furthermore, it demonstrated simplicity in computation and effectiveness in a broad range of experiments shown in our paper.

[1] Arora, Sanjeev, et al. "Generalization and equilibrium in generative adversarial nets (gans)." ICML, 2017.

[2] Bai, Yu, Tengyu Ma, and Andrej Risteski. "Approximability of discriminators implies diversity in GANs." ICLR, 2018.

[3] Berthelot, David, Thomas Schumm, and Luke Metz. "Began: Boundary equilibrium generative adversarial networks." arXiv preprint arXiv:1703.10717 (2017).

[4] Ham, Hyungrok, Tae Joon Jun, and Daeyoung Kim. "Unbalanced gans: Pre-training the generator of generative adversarial network using variational autoencoder." ICML (2020).

[5] Li, Shaojie, et al. "Revisiting discriminator in GAN compression: A generator-discriminator cooperative compression scheme." NeurIPS (2021).

[6] Liu, Shiwei, et al. "Don’t be so dense: sparse-to-sparse gan training without sacrificing performance." IJCV (2023).

[7] Chen, Tianlong, et al. "Data-efficient gan training beyond (just) augmentations: A lottery ticket perspective." NeurIPS (2021).

[8] Kang, Minguk, Joonghyuk Shin, and Jaesik Park. "Studiogan: A taxonomy and benchmark of gans for image synthesis." arXiv preprint arXiv:2206.09479 (2022).

---

### Decision · Program_Chairs · 2023-09-21

**Decision:**

Accept (poster)

**Comment:**

1x A, 2x BA, and 1x BR. This paper proposes a balance ratio metric between the generator and discriminator in sparse GAN training, and dynamically adjusts the sparsity of discriminator to improve the balance of GAN training. Most reviewers agree on accepting the paper due to its (1) clear presentation, (2) insightful motivations, (3) valuable topic, and (4) useful metric technique. The rebuttal has addressed the BR reviewer’s concern although his final rating is lacking.